# One-Dimensional Nanomaterials in Resistive Gas Sensor: From Material Design to Application

**Ze Wang** [1] , **Lei Zhu** [1,2,*] **, Shiyi Sun** [1] **, Jianan Wang** [1,3,*] **and Wei Yan** [1]

1   State Key Laboratory of Multiphase Flow in Power Engineering, Department of Environmental Science and Engineering, Xi'an Jiaotong University, 28 Xianning West Road, Xi'an 710019, China; wangze@stu.xjtu.edu.cn (Z.W.); kevin_sun23@stu.xjtu.edu.cn (S.S.); yanwei@xjtu.edu.cn (W.Y.)
2   School of Physics and Electrical Engineering, Weinan Normal University, Chaoyang Street, Weinan 714099, China
3   Zhejiang Research Institute, Xi'an Jiaotong University, 328 Wenming Road, Hangzhou 310000, China
*   Correspondence: cookkai@stu.xjtu.edu.cn (L.Z.); wangjn116@xjtu.edu.cn (J.W.)

**Abstract:** With a series of widespread applications, resistive gas sensors are considered to be promising candidates for gas detection, benefiting from their small size, ease-of-fabrication, low power consumption and outstanding maintenance properties. One-dimensional (1-D) nanomaterials, which have large specific surface areas, abundant exposed active sites and high length-to-diameter ratios, enable fast charge transfers and gas-sensitive reactions. They can also significantly enhance the sensitivity and response speed of resistive gas sensors. The features and sensing mechanism of current resistive gas sensors and the potential advantages of 1-D nanomaterials in resistive gas sensors are firstly reviewed. This review systematically summarizes the design and optimization strategies of 1-D nanomaterials for high-performance resistive gas sensors, including doping, heterostructures and composites. Based on the monitoring requirements of various characteristic gases, the available applications of this type of gas sensors are also classified and reviewed in the three categories of environment, safety and health. The direction and priorities for the future development of resistive gas sensors are laid out.

**Keywords:** resistive gas sensor; 1-D nanomaterial; materials design; gas detection; sensing mechanism; environment monitoring

## 1. Introduction

A variety of toxic and harmful gases generated during human activities (chemical production, interior decoration, food industry, medical treatment, transportation, etc.) have severely negative impacts on air quality and human health [1–5]. Nowadays, gas detection tools include infrared absorbers [6], ion mobility spectrometers [7], gas chromatographs [8], etc. However, these methods suffer from their time-consuming and complex operation, and their expensive analytical techniques, which make it difficult to meet the ever-growing demand [9,10]. Therefore, it is imperative to develop the next generation of detection techniques for toxic and harmful gases so as to protect humans from the potential health risk [4,11,12]. Gas sensors are advantageous over traditional detection methods because of their convenient operation, low cost, small size, low detection limit, etc. [13], which can convert the concentration signals of the target gases into visual signals to achieve fast and accurate gas detection for early warning. For these reasons, exploring and developing high-performance gas sensors has aroused extensive interest from scientists in recent years [14].

To date, many types of gas sensors with different transduction forms have been developed, e.g., catalytic combustion gas sensor [15], electro-chemical gas sensor [16], quartz crystal microbalance (QCM) gas sensor [17], thermal conductivity gas sensor [18], infrared absorption gas sensor [19] and resistive gas sensor [20,21]. Catalytic combustion

gas sensors make the combustible gas burn under the effect of catalysis, and then collect the resistance value of the temperature change as the output signal [22,23]. Electro-chemical gas sensors seek to determine the gas concentration through the conversion of electrical signals after the target gas has been oxidized/reduced [24]. Thermal conductivity gas sensors can convert signals related to the type and concentration of the gas into electrical signals [18]. Infrared absorption gas sensors measure the gas concentration via the specific infrared absorption spectra of different target gases, enabling qualitative and quantitative detection simultaneously [25]. QCM gas sensors are a kind of sensitive monitoring instrument with a selective adsorption film coated on the surface of the crystal [26]. Resistive gas sensors can effectively transform the gas change around the gas-sensing medium into resistance signals, thus achieving the goal of gas sensing [27–29]. An ideal gas sensor must have high responsivity, good selectivity, fast response/recovery, great stability/repeatability, and low costs for practical application. Hence, six-axes spider-web diagrams for evaluating the gas-sensing properties of various sensors are shown in Figure 1 [10,15–21,24].

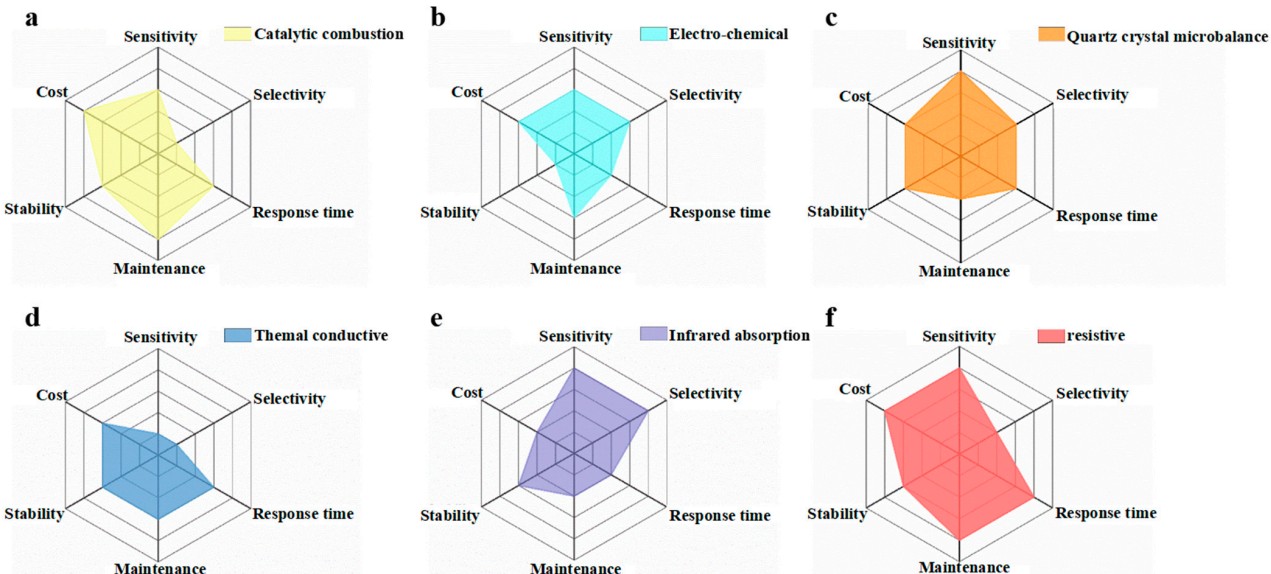

**Figure 1.** Six-axes spider-web diagrams for evaluating the gas-sensing properties of different sensors. (**a**) Catalytic combustion; (**b**) electro-chemical; (**c**) quartz crystal microbalance; (**d**) thermal conductive; (**e**) infrared absorption; (**f**) resistive gas sensors [10,15–21,24].

Compared with other gas sensors, the resistive gas sensor exhibits attractive advantages, including higher sensitivity, outstanding stability, flexibility (easy to operate and easy to utilize/integrate into portable devices), low power consumption, and lower operation cost, highlighting its great potential in gas detection, such as for environmental, safety and health monitoring [20,21,30–34]. However, the poor gas selectivity and high operating temperature of resistive gas sensors hinder their commercial application.

To develop high-performance resistive gas sensors, it is important to understand their gas-sensing mechanism as well as choosing suitable gas-sensing materials. The physical and chemical properties of sensing materials, such as dimension, morphology, structure, crystallinity, specific surface area, content of active sites, etc., play vital roles in the adsorption of the target gases, the electron transport and the chemical reaction rates, consequently affecting the performance of gas sensors [35–39]. Recently, one-dimensional (1-D) nanomaterials have become a focus for researchers. They have large surface areas, abundant exposed active sites and high length-to-diameter ratios, hence enabling fast charge transfers and efficient gas-sensitive reactions, as well as greatly enhancing the sensitivity and response speed of resistive gas sensors.

Hence, this article firstly concentrates on the configuration type and sensing mechanism of resistive gas sensors, and further compares the characteristics of various dimen-

sional nanomaterials applied in resistive gas sensors. Furthermore, this paper presents a comprehensive review of the recent research efforts and developments of 1-D nanomaterials as sensing materials used in resistive gas sensors, referring to the design and optimization of 1-D nanomaterials in resistive gas sensors and their potential application in the various fields of environment, safety and health monitoring (Figure 2).

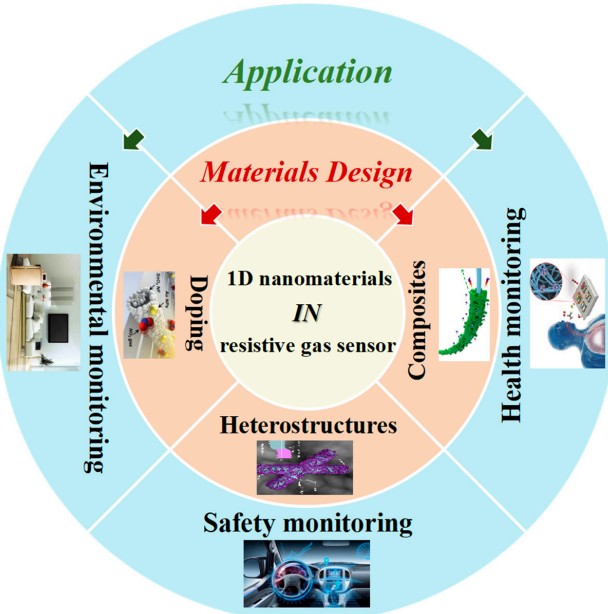

**Figure 2.** The design strategies of one-dimensional (1-D) nanomaterials and the applications of resistive gas sensors based on 1-D nanomaterials.

## 2. Introduction to Resistive Gas Sensor

To comprehensively study a resistive gas sensor, this review is carried out from four perspectives: performance index, configuration, sensing mechanism and 1-D nanomaterials.

### 2.1. Performance of Resistive Gas Sensor

The performance of resistive gas sensors can be generally quantified by the following five specific parameters [22,40]: (1) Sensitivity. Sensitivity is the smallest volume concentration of the target gas that can be sensed at the time of detection, which can be defined as $R_a/R_g$ for the reducing gases or $R_g/R_a$ for the oxidizing gases. $R_a$ is the resistance of the gas sensor in the reference gas (usually air), and $R_g$ stands for the resistance of the gas sensor in the target gas. A higher value of response means a higher sensitivity of the gas sensor to the target gas. (2) Selectivity. Selectivity is the ability of gas sensors to preferentially detect specific gases while avoiding the influences of other interfering gases. Selectivity is often evaluated according to the sensitivity ratio between the target gas and interference gas. A higher ratio means higher selectivity. (3) Response/recovery time. The time it takes for the sensor to achieve 90% of the total resistance change during adsorption and desorption is defined as response time and recovery time, respectively. Short response/recovery time is necessary for real-time gas monitoring. (4) Stability. Stability refers to whether the sensing response value remains unchanged after repeated use at a specific gas concentration. Sensors with good stability can reduce the influences of environmental interference factors, such as temperature and humidity, thus getting more reliable and accurate data. (5) Maintenance. Maintenance refers to whether the sensing response value remains unchanged during long-term operating at a specific gas concentration. Insufficient maintenance means that the sensor is no longer reliable and accurate after long-term operating. The difference between stability and maintenance is that the former is obtained by multiple repeated measurements under the same conditions, while the latter evaluates the stability of the sensor during long-term operation.

### 2.2. The Configuration Type of Resistive Gas Sensor

In terms of the configuration type, resistive gas sensors can be further classified as tubular, plate-like and flexible/wearable sensors. At present, the ceramic tube type is the dominant type of resistive gas sensor, mainly due to its developed preparation process and low cost [41–44]. As shown in Figure 3a, the ceramic tube consists of four sections including a ceramic tube, a Ni–Cr heater, Au electrodes and Pt wires. The heater is placed into the ceramic tube to provide the working temperature, and the comb-shaped Au electrodes are set outside the tube with the sensing materials coated on it. The sensing layer is deposited on the tubular insulator materials (generally a ceramic tube) to provide the gas signals.

In the plate-like configuration, a sensing layer (generally based on metal oxides or composites) is deposited over an insulating substrate (SiO$_2$ or Al$_2$O$_3$) with the interdigital electrodes (IDEs) in its front side [45–47]. The IDE sensors are favored by their miniaturization, fast response and low-cost mass fabrication, and can be directly applied in many areas (biomedical, environmental, industrial, etc.) without any design changes [48,49].

Flexible/wearable electronic sensors have attracted tremendous attention in recent years [50–56]. Flexible gas sensors are crucial components in portable electronic devices, benefiting from their small size, light weight, excellent mechanical properties, and ability to provide ultra-selective sensitivity in real-time analysis for environmental and health monitoring [57–59]. The structures and physical maps of various sensors are shown in Figure 3.

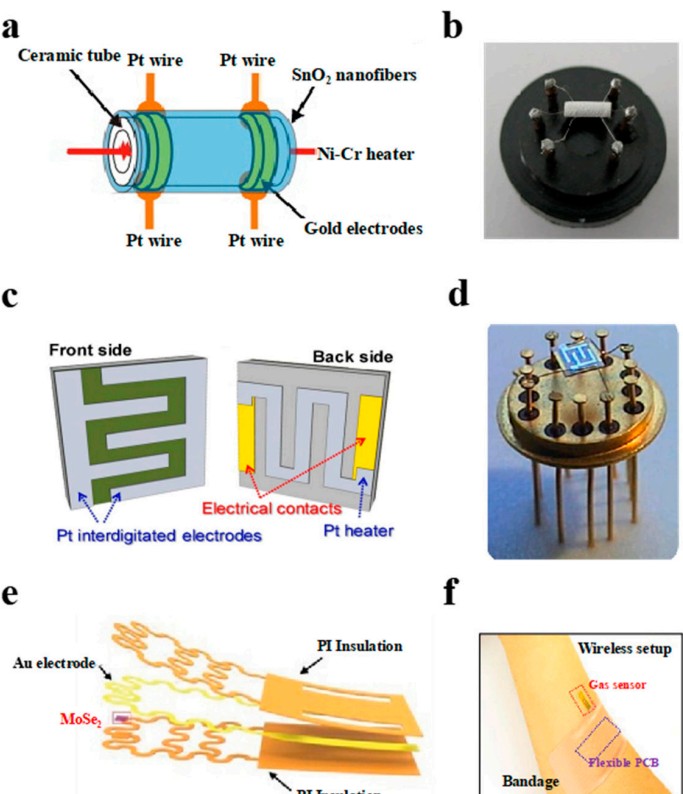

**Figure 3.** Schematic and optical images of various resistive gas sensor configurations: (**a**,**b**) tubular [60]; (**c**,**d**) plate-like [61]; (**e**,**f**) flexible wearable [62].

### 2.3. Sensing Mechanism of Resistive Gas Sensor

The gas sensing mechanism of a metal oxide semiconductor (MOS)-based resistive sensors, for example, is predominantly based on the changes in their resistance after they are exposed to the target gases due to the chemical interaction between target gas molecules and the adsorbed oxygen ions on the surface of the MOS. There are two main

sensing mechanisms, which are the surface charge layer model and the grain boundary barrier model.

The surface charge layer model involves lattice defects on the surface of the gas sensor material caused by a certain proportion of electron acceptors and donors. In an oxygen atmosphere, a certain amount of oxygen molecules will be adsorbed on the surface of the sensing material. The adsorbed oxygen molecules extract electrons from the conduction band and trap the electrons on their surface as ions, such as $O_2^-$, $O^-$, and $O^{2-}$, forming different surface energy levels. As a result, a space charge layer (p-type MOSs) or electron depletion layer (n-type MOSs) is formed. In a target gas atmosphere, the donor or acceptor gaseous electrons are adsorbed onto the metal oxide surface and exchange electrons with the MOS sensors, resulting in a change in the conductivity of the sensing material. The conductivity of the MOS gas sensor depends on the charge transfer mechanism in between the adsorbed gaseous species and MOSs as well as the analyte gas surface reaction. Figure 4a,b show the charge transfer on the surface of n-type and p-type MOS gas sensor [63,64].

The grain boundary barrier model is based on the energy band theory, which holds that the MOSs used as gas sensor materials are crystals composed of many crystal grains. These interconnected grains form larger aggregates connected to each other by grain boundaries. For n-type materials, when oxidizing gas (such as $O_2$) is adsorbed at the grain boundary, the adsorbed oxygen can combine with electrons on the material's surface to increase the surface electron barrier, further creating a large amount of positive charge in the material interface. As a result, a space charge layer will be formed between the negative charge and the positive charge, and result in a grain boundary potential barrier. When electrons in the conduction band transfer from one grain to another, the grain boundary potential barrier must be overcome [33,65,66].

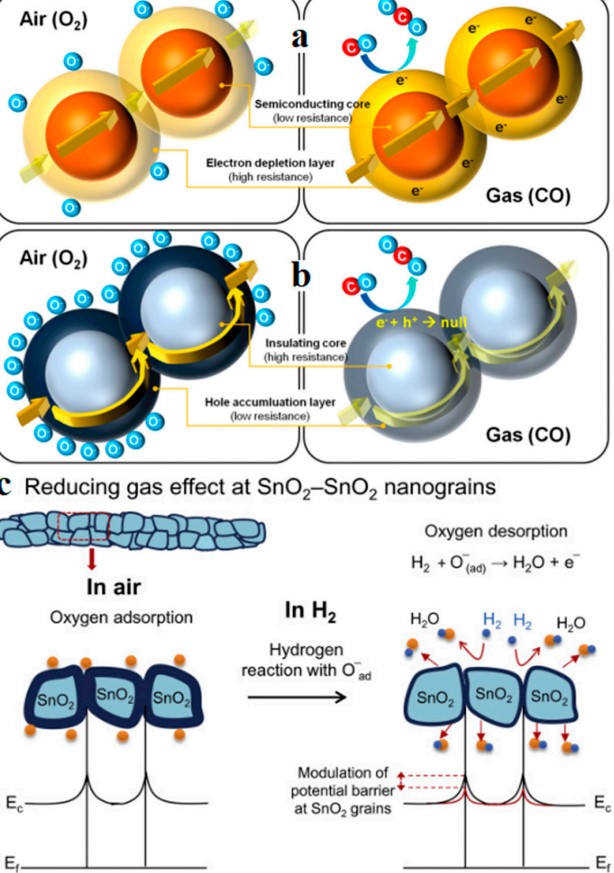

**Figure 4.** Simplified gas-sensing mechanism of a space-charged model of (**a**) n-type and (**b**) p-type semiconductor metal oxides [63]; (**c**) schematic diagram of the proposed bifunctional sensing mechanism: reducing gas ($H_2$) effect at $SnO_2$ homointerfaces [67].

The height of the barrier is directly proportional to the concentration of the adsorbed oxygen. A high concentration of adsorbed oxygen leads to a high potential barrier, and causes a high resistance in the sensing material. Once the adsorbed oxygen combines and reacts with the reducing gas, the electrons will be released and return to the conduction band. As a result, the space charge layer will disappear and the height of the potential barrier will decrease, thus leading to resistance change in the gas-sensing material. The grain boundary barrier model can account for the phenomenon whereby gas sensor resistance decreases in the reducing gas. For example, when $SnO_2$ is exposed to air, oxygen molecules will diffuse to the surfaces of $SnO_2$ nanocrystals, and obtain electrons from the conduction band to form oxygen ions, further inducing band bending between nanograins and formed depletion layers, consequently restricting the electron flow (Figure 4c) [67].

The gas-sensing mechanism of conductive polymer-based resistive sensors predominantly depends on the changes in resistance after they are exposed to the target gases, due to the chemical interaction or weak interaction between the target gas molecules and the polymers [68]. In terms of different target gases, there are three main chemical reactions, as follows: electron transfer ($NH_3$, $NO_2$, $I_2$, $H_2S$ and other reducing gases), local charge transfer (inert gas like CO, methanol, etc.) and acid–base reactions (acid gases such as HCL, $CO_2$, $H_2S$). Many important organic analytes, such as benzene, toluene and other toxic organic compounds, can hardly react with polymers under mild conditions. Therefore, it is difficult for conductive polymers to detect analytes by chemical reactions. However, there exist some weak physical interactions between these organic and conductive polymers, including adsorption, expansion of polymer matrixes, etc., which can affect the properties of polymers to achieve the detection of target gases.

The gas-sensing mechanism of carbon materials-based resistive sensors (take graphene as an example) is mainly based on their conductance changes upon the adsorption of sensing species [69]. Target gases with different structures and properties react with graphene in different ways. When graphene (a typical p-type semiconductor) is exposed to various gases, the response directions of its conductance are also different. The adsorption of electron-withdrawing gas molecules such as $NO_2$ can enhance the doping level of graphene and simultaneously increase its conductance. Conversely, electron-donating molecules, such as $NH_3$, de-dope graphene, which will decrease its conductance [70].

### 2.4. 1-D Nanomaterials

According to their dimension and morphology [39], nanomaterials can be divided into zero-dimensional (0-D) nanomaterials (nanoparticles [71], nanospheres [72], quantum dots [73], etc.), 1-D nanomaterials (nanorods (NRs) [74], nanotubes (NTs) [75], nanoneedles [76], etc.), two-dimensional nanomaterials (2-D) (nanomembranes [77], nanosheets [78], nanoplates [79], etc.) and three-dimensional (3-D) nanomaterials (nanoballs [80], nanoflowers [81], nanocubes [82], etc.). 0-D nanomaterials have more active sites, but the coarsening and agglomeration of 0-D nanomaterials will lead to insufficient specific surface, depleting the final sensing performance. The aggregation is no longer a problem when 0-D nanomaterials are deposited on a platform because of the interaction between the platform and 0-D nanomaterials [83]. Therefore, pure 0-D nanomaterials are unsuitable to for direct use as the gas-sensing material. As an alternative option, they are usually loaded in 1-D or 2-D materials to play the role of the active site for selectively catalyzing or absorbing target gases, consequently improving the selectivity and sensitivity of gas sensors. Recently, 1-D nanomaterials have been favored by researchers in the field of gas sensors due to their excellent physical, chemical and structural properties. They have larger specific surface areas compared to 0-D nanomaterials, as well as more exposed active sites distributed on their surfaces and interfaces. More importantly, the high length-to-diameter ratio of 1-D nanomaterials realizes a fast charge transfer, significantly enhancing the sensitivity and response speed, as well as reducing the operating temperature, of the gas sensors. However, when 1-D nanomaterials are integrated into a sensor device, they are always coated on the surface of the device first and then calcined, which can cause uneven coverage. In addition,

undesirable calcination times/temperatures will also affect the morphology and sensing performance of the materials. Hence, in terms of the sensor's micro-fabrication, the in-situ deposited 2-D MOS thin films using techniques such as chemical deposition, evaporation or sputtering are more advantageous [84–91]. 2-D nanomaterials have also drawn considerable attention in the gas-sensing field because of their high surface-to-volume ratio and modulated surface activities. Gas sensitivity performance will be significantly enhanced especially when the material is composed of porous sub-microstructures. However, the structures of 2-D nanomaterials are not as flexible as 1-D nanomaterials, because 1-D nanomaterials can be made into different structures, such as fibers, core–shells, hollow types, etc. Besides this, 1-D nanomaterials are more suitable for some applications that require the directional transmission of electrons. The 3-D nanomaterials employed in gas sensors are mostly composed of low-dimensional materials, possessing much better porous and mechanical properties than 1-D nanomaterials. Some of the 3D materials with hierarchical nanostructures, such as $SnO_2$ microsphere, exhibit high porosity and large specific surface areas beyond 140 $m^2$/g [92]. Some metal organic frameworks (MOFs) and 3-D graphene frameworks also have similar properties, exceeding those of 1-D nanomaterials [4,93]. 3-D structures have been proven to be more conducive to the transfer and storage of gas molecules, ensuring both high-flux surface reactions with gases and shorter electronic transfer paths. The morphology and characteristics of materials with different dimensions are further shown in Figure 5.

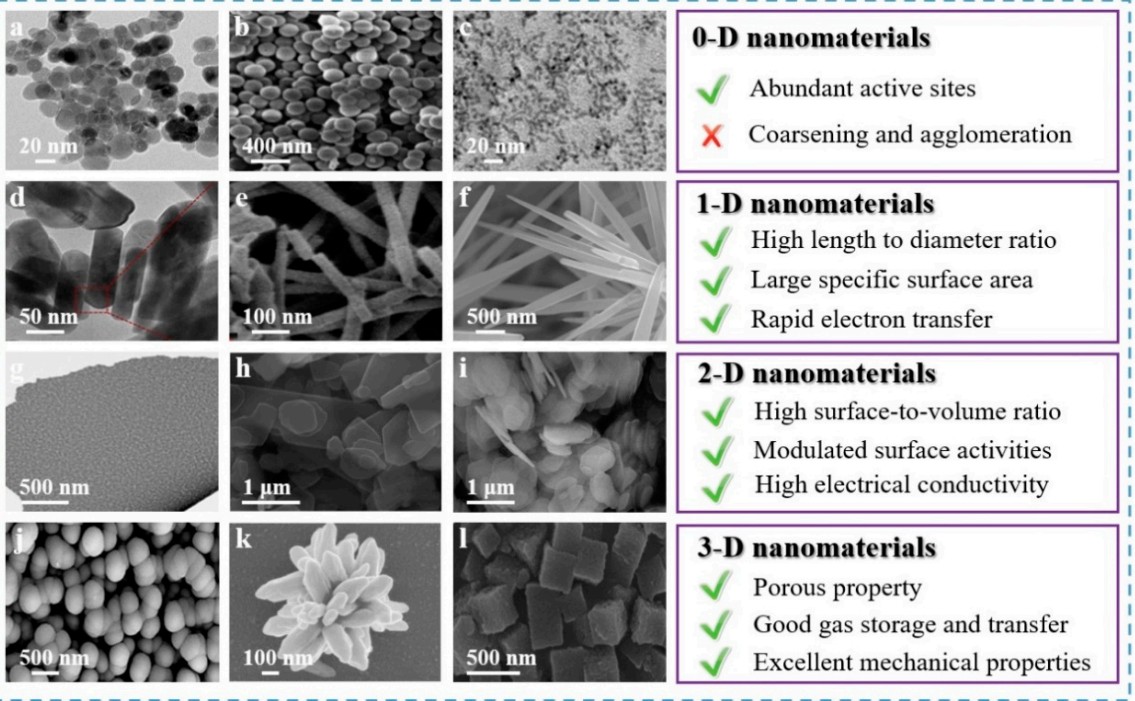

**Figure 5.** The morphologies of 0-D nanomaterials: (**a**) nanoparticles [71]; (**b**) nanospheres [72], and (**c**) quantum dots [73]; 1-D nanomaterials: (**d**) NRs [74]; (**e**) NTs [75], and (**f**) nanoneedles [76]; 2-D nanomaterials: (**g**) nanomembranes [77]; (**h**) nanosheets [78], and (**i**) nanoplates [79]; 3-D nanomaterials: (**j**) nanoballs [80]; (**k**) nanoflowers [81], and (**l**) nanocubes [82].

The use of 1-D nanomaterials in resistive gas sensors have been widely developed based various fabrication methods, [94] including template [95], electrospinning [96–99], hydrothermal [100,101] and sol-gel [102]. Depending on the processing route and treatment, different types of nanostructures with diverse surface morphologies can be achieved. 1-D nanomaterials can also be categorized into four different families: nano-long fibers (nanowires [94], hollow nanofibers [96], etc.), nano-short fibers (NRs [103], NTs [104], etc.), core–shell fibers [30,105], and yolk–shell fibers [99,106,107]. Some 1-D nanomaterials and their fabrication schematic illustrations are shown in Figure 6.

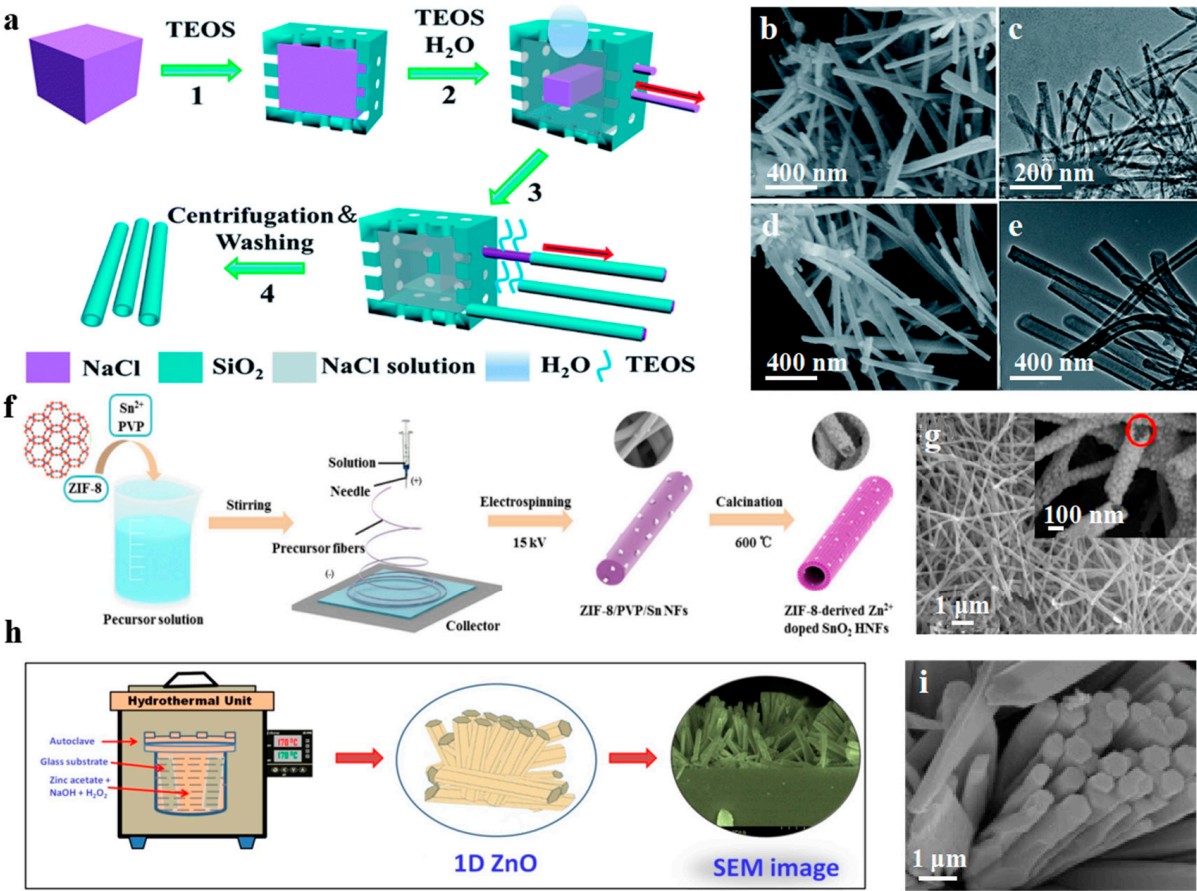

**Figure 6.** (**a**) Schematic illustrations of SiO$_2$ nanowires and hollow SiO$_2$ NTs constructed by the template method; (**b**) SEM and (**c**) TEM images of SiO$_2$ nanowires; (**d**) SEM and (**e**) TEM images of hollow SiO$_2$ NTs [95]; (**f**) schematic illustrations of SnO$_2$ hollow nanofibers (HNFs) constructed by electrospinning method; (**g**) SEM image of SnO$_2$ HNFs [96]; (**h**) schematic illustrations of ZnO NRs constructed by hydrothermal method; (**i**) SEM image of ZnO NRs [101].

## 3. Materials Design

Conductive polymers [68], MOSs [66], carbon materials (graphene, reduced graphite oxide, carbon nanotubes, etc.) [69] and organic/inorganic compounds have been successfully used as sensing materials in resistive gas sensors. However, 1-D nanomaterials alone struggle to meet the requirements of high sensitivity, rapid responses, and high selectivity. Hence, several effective modification strategies have been developed for 1-D nanomaterials to help them achieve the aforementioned objectives, as follows: (1) Doping. The doping of general metals and noble metals into the lattice of metal oxide will result in lattice mismatch, and may provide more adsorption sites and charge carriers, thereby improving the sensitivity of gas sensor. (2) Heterostructures. A heterojunction is defined as the interface between two dissimilar semiconductors (one is the host, and the other is the guest) that form a junction (n-n, p-p or p-n) linked with an energy band structure due to the alignment of their fermi level. The heterojunction can improve the sensor performance through facilitating catalytic activity, increasing adsorption, and creating a charge carrier depletion layer to produce greater modulation in its resistance. (3) Composites. Synthesizing a composite with two or more different substances can overcome the limitations of individual materials, and improve the intrinsic sensor signal via synergistic effects.

### 3.1. Doping

Doping includes general metal doping (Al, Ga, Ca, Zr, etc.) and noble metal doping (Au [108], Ag [109], Rh [110], Pt [111], etc.). General metal doping can change both the adsorption sites and diffusion paths of gases and the energy band structures of materials

by optimizing the grain size, which is widely considered to be a simple and effective way to elevate the performance of gas sensors [112–114]. Chen et al. [115] confirmed that different heteroatom substitutions can modulate the Fermi level of $In_2O_3$ nanofibers (NFs) and significantly influence the formaldehyde response. Especially, Al-doped $In_2O_3$ NFs exhibit a higher Fermi level than other $In_2O_3$-based sensing materials, hence achieving a much higher response (60) to formaldehyde (100 ppm) at the low operating temperature of 150 °C, along with a fast response speed (2–23 s) and improved selectivity. However, except for the effective dopants (Al, Ga and Zr), other dopants (e.g., Ti, V, Cr, Mo, W and Sn) have been proven to reduce the original Fermi level of $In_2O_3$, going against the sensing response. These results indicate that the Fermi level has an important effect on gas-sensing properties. Zhao et al. [104] synthesized Ca-doped $In_2O_3$ NTs via electrospinning and annealing technology, confirming that Ca dopants play a key role in the crystal structure, grain size, morphology, and oxygen vacancy concentration of the material (Figure 7a,b). The average size of $In_2O_3$ grains decreased initially following an increase in Ca content from ≤3 mol% to ≥7 mol%. In particular, the sensor based on 3% Ca-$In_2O_3$ NTs showed the best sensing performance toward ethanol, with high sensitivity, good selectivity, long-term stability and excellent reproducibility (Figure 7c,d). Hence, in terms of general metal atom doping, its enhanced sensing performance can mainly be attributed to the positive effect of grain size, oxygen vacancy concentration and catalytic activity (Figure 7e).

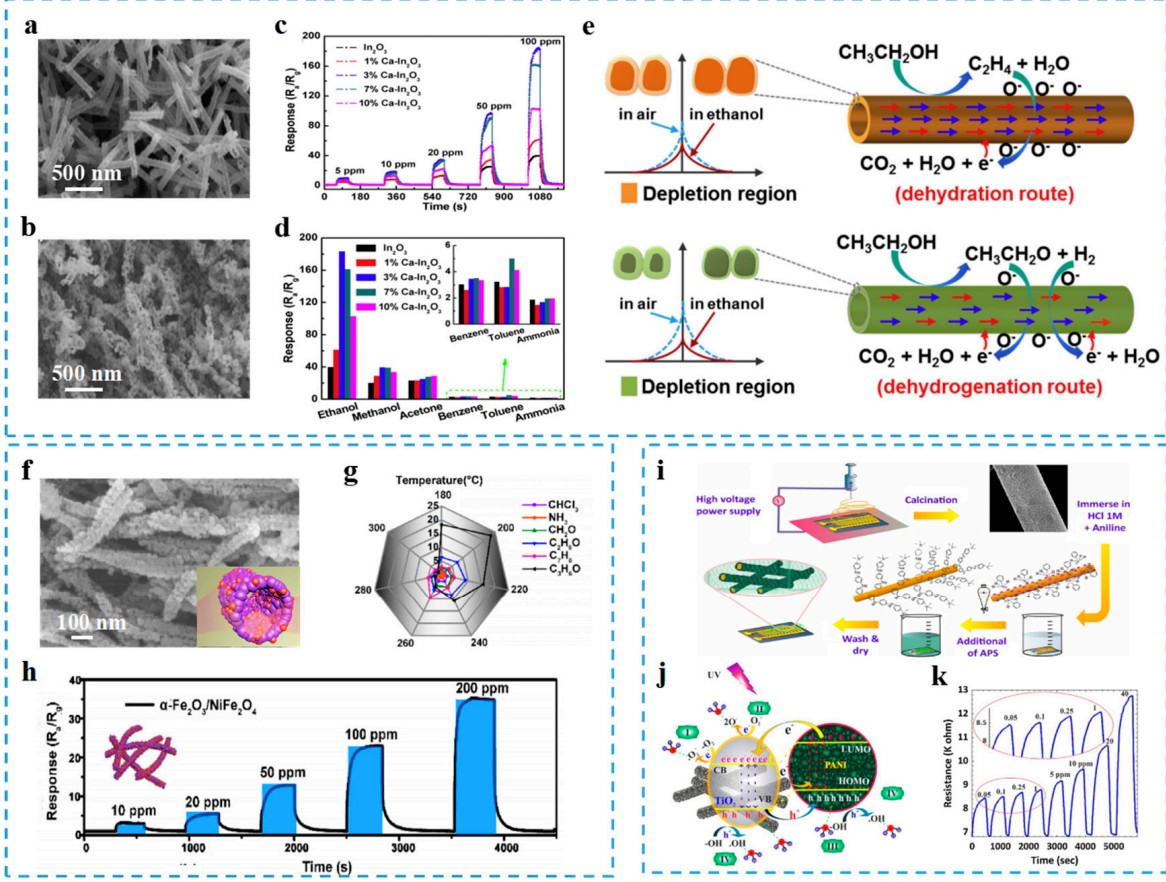

**Figure 7.** SEM images of (**a**) 3% Ca−$In_2O_3$ NTs and (**b**) 10% Ca−$In_2O_3$ NTs; (**c**) the corresponding response–recovery curves and (**d**) selectivity of the Ca−$In_2O_3$ sensor to various gases (100 ppm) at 240 °C; (**e**) schematic diagrams of pure $In_2O_3$ NTs and Ca−$In_2O_3$ NTs exposed to air and ethanol, respectively [104]; (**f**) FESEM image of $\alpha$−$Fe_2O_3$/$NiFe_2O_4$ NTs; (**g**) polar graphs of selectivity to different gases and (**h**) response transients (200 °C) to 10–200 ppm acetone of $\alpha$−$Fe_2O_3$/$NiFe_2O_4$ sensor [116]; (**i**) preparation of $TiO_2$/PANI composites; (**j**) $NH_3$ absorption on $TiO_2$/PANI upon exposure to UV light; (**k**) real-time resistance change with different $NH_3$ concentrations [105].

As for noble metal doping, when noble metal dopants meet MOSs, a charge transfer will occur between the materials and electrons and accumulate in the noble metal, thus balancing the Fermi level, and further forming a depletion layer on the metal oxide. Thus, an enhanced sensing response can be achieved by increased initial material resistance [117]. Additionally, noble metal nanoparticles, as excellent active catalysts, are of great benefit to the adsorption of oxygen molecules, and increase the concentration of oxygen ions, hence improving the sensitivity and response speed of the materials. The target gas molecules are directly adsorbed into the noble metal nanoparticles, then migrate to participate in the reaction. This spillover effect can significantly improve gas sensing performance. For gas sensors based on metal oxides, room- and low-temperature sensing have attracted enormous interest given their low power consumption [84,85,118–121]. Noble functionalization on metal oxides can also lower the operating temperature of gas sensors [15,112,122–126]. Kou et al. [110] reduced the grain size of $SnO_2$ by Rh-doping, and further induced changes in electron concentration and oxygen component distribution, improving the response of a Rh-doped $SnO_2$ gas sensor toward acetone at the lower working temperature of 200 °C. Yu et al. [111] synthesized a Pt-ZnO nanorod array, which was directly grown in-situ on a seeded alumina tube sensor platform. Pt-ZnO-based sensors exhibited high responses to 23.1–100 ppb $H_2S$ at 260 °C—5.8 times higher than the ZnO sensor. They also showed a limit of detection of 1.1 ppb, and outstanding selectivity to $H_2S$ against interfering gases ($NH_3$, $NO_2$, $H_2$, ethanol, acetone). The excellent sensing performance may be attributed to the synergistic effects of the catalytic and electronic promotion of Pt.

Although doping is one of the effective ways to improve the sensing performance, some dopants and unsuitable doping contents will also adversely affect the sensing performance [115]. Excessive doping will cause the agglomeration of doped particles, reduce the specific surface area of the material, reduce the active sites, and affect the gas-sensing reaction.

### 3.2. Heterostructures

Pure MOSs struggle to satisfy all the requirements of a high-performance gas sensor. Therefore, constructing a heterojunction by combining two or more MOSs is an alternative strategy to realize a synergic effect, as it can effectively enhance the gas-sensing performance by introducing more electron depletion layers, changing the energy band structure, increasing adsorption sites, or boosting the catalytic activity of the materials, thus acquiring high sensitivity and favorable selectivity [43,127]. According to the conductivity types of MOSs, the heterojunction structures can be divided into n–n, p–p and p–n heterojunctions. When p-type and n-type MOSs are combined, the material resistance will increase with the reduction in electrons at the p–n heterojunction interface. For the n–n heterojunction, electrons will transfer from the high-Fermi level material to the low-Fermi level material at the n–n heterojunction interface, where the MOSs with a high Fermi level create a depletion layer and the MOSs with a low Fermi level form an accumulation layer. The accumulation layer is depleted by the continuously absorbed oxygen on the surface, further narrowing the conduction band and improving the sensing response. The n–n heterojunction might result in increased material resistance due to the electron confinement induced by the different work functions, and this enhances the response upon exposure to gaseous molecules. [128]. For the p–p heterojunction, holes as the main carriers in p-type MOSs will transfer from a high-Fermi level material to a low-Fermi level material at the interface. The former makes a hole depletion zone, while the latter forms a hole accumulation zone [64].

The formation of these heterojunctions can not only effectively accelerate electron transport, but can also enhance oxygen adsorption and form abundant oxygen vacancies on the surface of the material, serving as new reaction active sites [117,128–130]. In actuality, some heterojunctions can also be introduced in the doping process of metal atoms. Under high doping concentrations, besides entering into the lattice to replace the original metal atoms, the excessive doped atoms are also able to nucleate and further form new oxide phases, thus creating a heterojunction in the original MOSs to synergistically enhance the gas-sensing performance. Huang et al. [127] synthesized one-dimensional $NiS$–$In_2O_3$ com-

posites by electrospinning and calcination techniques. Benefiting from the superiorities of the created p–n heterojunction, NiS–$In_2O_3$ exhibited enhanced sensitivity, high selectivity and outstanding stability toward ethanol; Park et al. [131] synthesized porous $SnO_2$–CuO hollow nanofibers (HNFs) via a two-step process of single-needle electrospinning and thermal treatment, further achieving the construction of p–n heterojunctions and an enhanced surface area. The gas sensor assembled form $SnO_2$–CuO NTs achieved high/fast responses at low operating temperatures for $H_2S$ gas. Zhou et al. [116] synthesized $\alpha$-$Fe_2O_3$/$NiFe_2O_4$ NTs via a simple hydrothermal route using MOF as the sacrificial templates (Figure 7f). Benefiting from the formation of an $\alpha$-$Fe_2O_3$/$NiFe_2O_4$ heterostructure, the charge separation and carrier depletion layer could be significantly enhanced at the $NiFe_2O_4$/$\alpha$-$Fe_2O_3$ interfaces, leading to enhanced conductance modulation. A gas sensor based on $\alpha$-$Fe_2O_3$/$NiFe_2O_4$ NTs exhibited outstanding sensitivity (23 to 100 ppm acetone) and selectivity (23 to acetone compared with <8 to others) at low operating temperatures (200 °C) (Figure 7g,h).

From the above cases, we can see that the construction of heterostructures is one of the best approaches to effectively enhance MOS nanostructures. However, heterostructure design in some cases leads to increases in operating temperature due to the higher activation barrier at the interface [132,133].

### 3.3. Composites

The main composites employed in resistive gas sensors can be divided into two categories: the conductive polymers combined with MOSs and the carbon materials combined with polymers, MOSs or metals. Building a composite system can realize synergic and complementary effects to enhance the specific physical/chemical properties of gas-sensing materials, such as electrical conductivity, mechanical strength, thermal stability, etc.

Conductive polymers are a distinctive group of organic materials that exhibit the electrical and optical properties of both metals and semiconductors, which can be further doped to a conductive state. Various conductive polymers such as polyaniline (PANI), polydiphenylamine (PDPA), polypyrrole (PPy), and polythio-phene (PTh) have been reported for use in $NH_3$ gas sensors due to their excellent electronic properties, low operating temperatures and intrinsic redox reactivity. However, pure conductive polymers still suffer from their poor stability and low processing ability. In order to solve the above-mentioned problems, a range of techniques, including metal catalyst-doping, compositing with MOS, and carbon series nano-materials (carbon nanotubes or graphene), have been proposed. Among these, many studies on conductive polymers–MOS composites have been reported. Seif et al. [105] used electrospun $TiO_2$ NFs as the core material and PANI as the shell material to prepare $TiO_2$/PANI core–shell nanofiber film $NH_3$ sensors (Figure 7i). In this system, $TiO_2$ NFs first adsorbed the oxygen molecule on their surface, and the absorbed oxygen molecules captured electrons near the surface of the $TiO_2$. By introducing $NH_3$ gas, the gas molecules would further pass through the porous PANI shell and interact with these oxygen ions, and then return the absorbed electron to the $TiO_2$ (Figure 7j). The synthesis of PANI/$TiO_2$ core–shells is dominated by p-type PANI shells rather than n-type $TiO_2$. Due to the greater number of absorption sites and the synergetic effect between $TiO_2$ and PANI, the core–shell composites displayed outstanding sensitivity, a low detection limit (as low as 50 ppb under high humidity conditions), and excellent selectivity toward various volatile organic compounds (Figure 7k).

In addition, Li et al. [134] fabricated $SnO_2$/PPy composite NFs via gas phase polymerization, and further studied their gas sensitivity to low-concentration $NH_3$ gas at room temperature. $SnO_2$/PPy composite NFs have a higher responsiveness to $NH_3$ compared with pure PPy NFs. When the $NH_3$ concentration was 0.001‰ ~0.0107‰, the sensitivity of the $SnO_2$/PPy composite NFs was 6.2 ppm and the lowest detectable $NH_3$ was 0.257‰, with a fast response/recovery ability and high repeatability. This outstanding sensing performance of $SnO_2$/PPy composite NFs was mainly attributed to the high specific surface

area and the cross-arranged nanostructure of the composite, which could facilitate the free flow and diffusion of $NH_3$ molecules and provide more reaction sites.

Carbon materials sense gas mainly via the detection of conductivity changes in materials caused by gas adsorption [21]. Carbon materials possess large theoretical specific surface areas, and hence can provide a large sensing area for the adsorption of gas molecules [65]. Besides, they also exhibit high carrier mobility and low resistance at room temperature [18,135]. However, using pure carbon materials as sensing materials still has practical problems, such as poor reversibility. Therefore, various carbon-based composites have been developed and applied to gas sensors to optimize the sensing performance factors of stability, repeatability and response/recovery time.

Carbon/polymer composites have a porous microstructure, which can accelerate the adsorption and diffusion of gas molecules, promote the conductivity change rate of the sensing layer, and improve the gas sensitivity. Luo et al. [136] reported a flexible fabric gas sensor based on the rGO-PANI/cotton thread nanocomposites fabricated via the in-situ polymerization technique for the detection of sub-ppm-level $NH_3$ gas. These sensors exhibited a fast response (122 s), high sensitivity (6 to 100 ppm $NH_3$), remarkable long-term stability (the response slightly decreased from a value of 6 after 30 days) and high selectivity (6 to $NH_3$ compared with 1.01–1.3 to others), owing to both the 1-D distinctive nanostructure and the synergistic effect between rGO nanosheets and PANI. This elaborate sensor with low power consumption is considered to be a potential flexible electronic device for the further detection of $NH_3$ gas.

Carbon/MOS composites can also considerably improve the performance of gas sensors due to thier electronic structure modulation, large specific surface area, and heterostructure. Zheng et al. [137] synthesized novel $SnO_2$ NRs/ethylenediamine (EDA)-modified rGO composites via the one-step hydrothermal method. This $SnO_2$/EDA-rGO $NO_2$ sensor displayed a low detection limit (100 ppb), rapid response/recovery time (73/81 s) and excellent selectivity (282 to 1 ppm $NO_2$ compared with 5 to 20 ppm CO). This remarkable sensing performance is attributed to the synergy of the selective adsorption achieved by using modified EDA for the adsorption sites and the electronic structure modulation achieved by the $SnO_2$/rGO composites. Reddy et al. [138] fabricated $SnO_2$, Al–$SnO_2$ and GO–Al–$SnO_2$ NFs, and investigated their gas sensitivity toward $H_2$ gas. GO–Al–$SnO_2$ NFs have a higher response to $H_2$ than the other two materials. The response of GO–Al–$SnO_2$ NFs to 0.1‰ $H_2$ could reach 23.8 at a working temperature of 300 °C, with a response/recovery time of 2.2/1.4 s. The improved $H_2$ response mainly benefits from the smaller diameter (about 100 nm), larger specific surface area (29 $m^2$/g), better thermoelectric conductivity and greater quantity of oxygen vacancies of GO–Al–$SnO_2$ NFs resulting from the synergistic effects of the loaded graphene and Al.

Metal nanoparticles, such as Pt and Pd, fixed on carbon materials can catalyze the gas reaction and improve the gas response. Kim et al. [139] synthesized ternary nanocomposites containing rGO/metal (Pd or Pt)-coloaded $SnO_2$ NFs using the sol-gel method combined with electrospinning. This ternary composite showed a better gas sensing response than the $SnO_2$ and rGO/$SnO_2$ materials toward all the $C_6H_6$, $C_7H_8$ and CO gases. At a working temperature of 200 °C, the sensing responses of $SnO_2$ NFs, rGO/$SnO_2$ NFs and rGO/Pd/$SnO_2$ NFs to 1ppm $C_6H_6$ were 1.6, 3.3 and 8.3, respectively. The enhanced response of this rGO/metal-coloaded $SnO_2$ NF sensor was attributed to the high specific surface area, multiple heterojunctions among various materials, and the special sensitizing effects of Pt and Pd.

The features of the current carbon gas-sensing materials used in three major compositing systems can be summarized as follows: (1) conductive polymers can improve both the selectivity and sensitivity of carbon materials, but suffer from a relatively low stability. (2) MOSs improve the sensitivity of the carbon materials to a large extent, but with long response times and poor repeatability. Metal nanoparticles can effectively catalyze the gas-sensing reaction of carbon materials; however, they exhibit poor selectivity in the

mixed gas environment. Therefore, it is urgent to develop a comprehensive strategy to accelerate the practical application of carbon materials in resistive gas sensors.

## 4. Application

To date, resistive gas sensors based on 1-D nanomaterials have been widely employed in various fields, which can be mainly classified into three sections: environment, safety and health. The gases that need to be detected are obviously different in each field, thus leading to the different requirements for gas-sensitive characteristics of these 1-D gas-sensing materials, such as sensitivity, working temperature, stability, response/recovery rate, etc. Herein, resistive gas sensors based on 1-D nanomaterials for the three typical applications are further reviewed, as follows.

### 4.1. Environmental Monitoring

Traditional environmental monitoring technologies, such as mass spectrometry, gas chromatography, and optical measurement, make it possible to measure air pollutants with high precision. However, their wide application is still limited by the portability, high cost, complex operation and lack of real-time capability. In recent years, with the development of resistive gas sensors, personalized and localized environmental monitoring, rather than global and average monitoring, has received increasing attention. In the environmental field, the gases that must be monitored mainly include $NO_2$, $NH_3$, and some volatile organic compounds (VOCs, including formaldehyde (HCHO), methanol, toluene, etc.).

Man-made $NO_2$ (toxic and irritating) is mainly released by high-temperature combustion processes, such as vehicle exhaust and boiler exhaust. It is also one of the causes of acid rain, with lots of negative environmental effects. $NO_2$ can irritate the eyes and upper respiratory tract, and long-term inhalation of $NO_2$ can cause respiratory tract inflammation [140,141]. Lim et al. [142] prepared a transparent nanopattern of straight Au–$SnO_2$ NFs via direct-write and near-field electrospinning, which could detect $NO_2$ gas at room temperature under visible light illumination (Figure 8a). The extremely low coverage of sensing materials (about 0.3%) contributed to the high transparency (~93%) of the sensor, enabling the material to be completely exposed to the gas and thus promote the gas-sensing and photoactivation reactions (Figure 8b). In particular, the loaded Au particles further enhanced the response to $NO_2$ owing to the surface plasmon resonance effect of Au. The sensor exhibited an enhanced response (300) in sunlight to 5 ppm $NO_2$, a reproducible response to sub-ppm levels of $NO_2$, a detection limit as low as 6 ppb, and a high $NO_2$ response in both dry and relatively humid atmospheres (RH 50% and 70%) (Figure 8d,e). A gas sensor with high transparency and room-temperature operation is advantageous to promote the development of transparent electronic devices and smart windows wirelessly connected to the Internet of Things (IoT) (Figure 8c). Suh et al. [143] fabricated edge-exposed $WS_2$ synthesized on $SiO_2$ NRs, achieving highly sensitive and selective $NO_2$ detection. Its response to 5 ppm $NO_2$ reached 151.2 at room temperature, which is much higher than that to interference gases (3.44 to CO, 0.47 to $H_2S$). Its high performance is mainly attributed to its highly porous 1-D nanostructure and the chemically reactive edge sites of $WS_2$ serving as favorable active sites for the direct interaction with target gas molecules. The excellent gas-sensing performance of $WS_2$ NRs suggests its possible future pathways and prospective uses in gas sensors in IoT applications; Godse et al. [101] reported a simple, surfactant-free and template-less hydrothermal method for the fabrication of 1-D ZnO nanostructured sensing materials. The size and morphology of this ZnO nanostructure can be adjusted by modifying the temperature and time of the hydrothermal synthesis reaction, thereby optimizing the gas sensitivity parameters of the ZnO sensor. The ZnO sensor showed a significant response to 70 to 5 ppm $NO_2$, with a response time of 16 s. This material can easily realize large-scale preparation in gas sensors for future commercial applications.

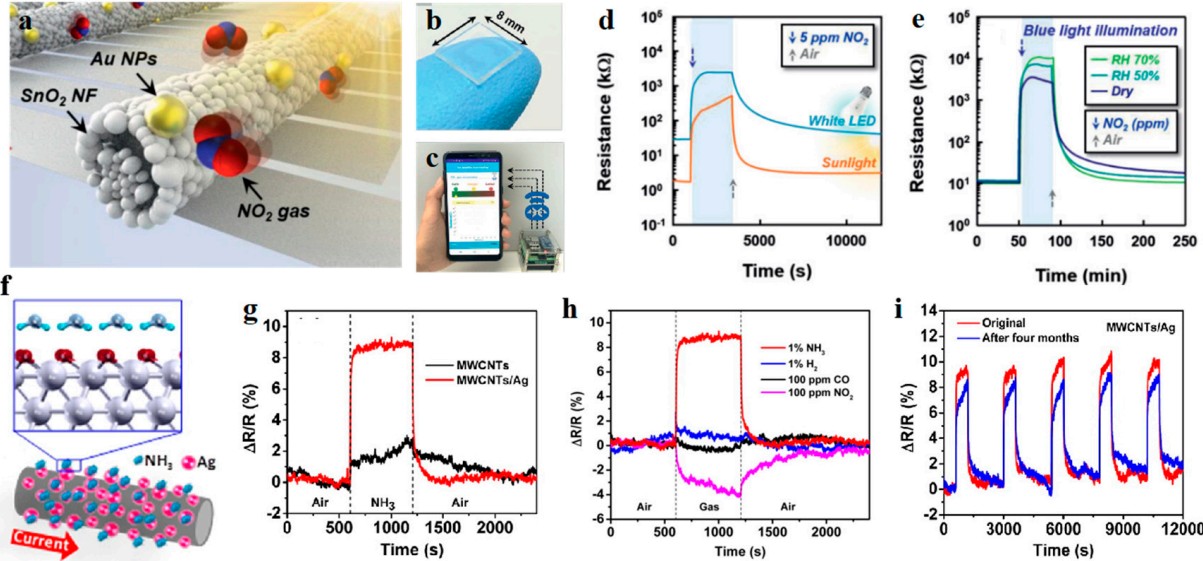

**Figure 8.** (**a**) Schematic illustration for Au−SnO₂ NFs; (**b**) photograph of transparent sensor; (**c**) wireless sensor module that can communicate with smart phones; (**d**) resistance change in reponse to 5 ppm NO₂ under white LED and sunlight, and (**e**) in different atmospheres [142]; (**f**) schematic illustration for Ag NC MWCNTs; (**g**) room−temperature dynamic sensing response before and after Ag NCs decoration; (**h**) comparison of sensing response to various gases; (**i**) comparison of sensing response toward 1% NH₃ before and after 4 months of storage in air [144].

NH₃ can burn skin, eyes, and the mucous membranes of respiratory organs. Excessive inhalation will cause lung swelling and even death [145,146]. The main NH₃ sources are the burning of fossil fuels, agriculture, animal husbandry, biological waste, and the release of indoor construction and decoration materials [147–149]. Kumar et al. [150] synthesized hexagonal-molybdenum oxide (h-MoO₃) NRs via the facile chemical bath technique. The absorption of h-MoO₃ NRs was achieved from the 220 to 320 nm wavelength range, with a band gap of ~3.7 eV. The h-MoO₃ NRs NH₃ sensors demonstrated aa positive response to NH₃ (114 to 100 ppm at 200 °C), as well as long-term firmness (gas response altering a bit in ~30 days). They show promise in industrial applications, such as in fertilizers, refrigeration industries, etc. Zheng et al. [151] constructed a novel 1-D core–shell heterostructure via a layer-by-layer synthesis method. This heterostructure has a single copper phthalocyanine (CuPc) ribbon core coated with a uniform isoreticular metal–organic framework-3 (IRMOF-3) shell. Due to the integration and synergy of the CuPc cores and the IRMOF-3 shells, the gas sensors based on such single heterostructures exhibited high sensing properties (selectivity, sensitivity, stability and reusability) for NH₃ detection at about 60% relative humidity at room temperature. Even when the concentration ratio of NH₃ (5 ppm) and interfering gas (500 ppm) was 1:100, the response to NH₃ was 34–265 times higher than that to the interfering gas (methanol, ethanol, acetone, NO₂, NO). The selective adsorption capacity of the MOF shells plays a vital role in the selectivity of the sensors. This work provided new opportunities for the development of toxic gas safety control and environmental monitoring technology. Seifaddini et al. [152] synthesized graphene nanoribbons (GNRs) by unzipping CNTs based on the chemical oxidation method, and then introduced Au as a catalyst material in the GNR structure via the sputtering approach. The AuGNR and GNR sensors were tested at room temperature and showed 34% and 12.1% responses to 25 ppm NH₃, respectively. The higher response of the AuGNR sensor benefited from the excellent catalytic property of Au and the faster electron transfer. The AuGNR sensor also showed excellent selectivity to NH₃ among other gases (methanol, propanol, toluene and acetone). Cui et al. [144] synthesized Ag nanocrystal-functionalized multiwalled carbon nanotubes (Ag NC-MWCNTs) based on a simple mini-arc plasma method combined with an electrostatic force-directed assembly process (Figure 8f). The addition of Ag NCs to MWCNTs resulted in dramatically improved

sensitivity toward $NH_3$ gas (Figure 8g). The fully oxidized Ag surface plays a critical role in the response of the gas sensor. $NH_3$ molecules are adsorbed at Ag hollow sites on the AgO surface, with H pointing toward Ag. A net charge transfer from $NH_3$ to the Ag NC–MWCNTs hybrid causes a conductance change in the hybrid. The sensor also exhibited high selectivity to various gases and good long-term maintenance (Figure 8h,i). Some other $NH_3$ gas sensors have also been described above [105,134,136].

Methanol, as an important raw material and high-performance fuel, has been widely used in various fields [153]. Methanol vapor can damage human respiratory mucosa and eyesight. Methanol and air can form an explosive mixture, which may burn and explode when exposed to heat sources and open flames [154]. All these negative outcomes highlight the significance of methanol monitoring [155,156]. Liang et al. [157] prepared 1-D ZnO NFs via the hydrothermal method and grew porous large-surface area $NiCo_2O_4$ nanosheets on the surface of 1-D ZnO nanofibers via chemical bath deposition. This novel ZnO–$NiCo_2O_4$ core–shell heterostructure exhibited a high responsivity (1.96 to 5 ppm methanol) and high stability, whereas pristine $NiCo_2O_4$ and ZnO were almost nonresponsive at the same concentration. The enhanced gas sensing properties were mainly attributed to the unique core–shell heterostructure between $NiCo_2O_4$ nanosheets and ZnO nanofibers, implying the potential of these as-designed ZnO–$NiCo_2O_4$ core–shell p–n nanocomposites for high-efficiency methanol gas detection. Hu et al. [158] prepared Pd-doped $CeO_2$ NFs through an electrospinning and subsequent calcination method. The response of the 3%Pd–$CeO_2$ sensor was 4 times higher than that of the pure $CeO_2$ sensor, with rapid response/recovery time (1/3 s) and long-term stability (deviation less than 5% after a month), indicating that Pd doping is an effective strategy to enhance the methanol response of $CeO_2$. The improvement of gas sensitivity profited from the created active adsorption sites of PdO, the appropriate content of $Ce^{3+}$ ions adjusted by Pd doping, and the introduction of an n-type $CeO_2$/p-type PdO heterojunction. Sinha et al. [159] also synthesized a CNT/ZnO composite nanomaterial at low temperatures. This as-prepared nanocomposite changes from a p-type (low temperature) to an n-type (high temperature) when the temperature is above 150 °C. The CNT/ZnO sensor had a large operating temperature range of 50–250 °C, low power consumption and high selectivity. The response to methanol was 8 times higher than that to ethanol. Furthermore, only a super-low bias voltage ($\leq$10 mV) was required to drive the VOCs detection process, confirming the potential applicability of this sensor for future self-power-driven devices.

HCHO, as a well-known colorless and irritating carcinogenic gas, is unavoidably released from industrial chemical synthesis, interior decoration, and the incomplete combustion of fuel and tobacco, etc. In the medicine field, HCHO is also frequently used as an antiseptic and disinfectant [160–162]. Shin et al. [163] incorporated Pt single-atom (SAs)-anchored carbon nitride (MCN) nanosheets in a Sn-containing precursor and formed a 1-D nanofiber structure by electrospinning, further generating multi-heterointerface-engineered structures among Pt SAs, carbon nitride and $SnO_2$ (Pt-MCN-$SnO_2$) through calcination (Figure 9a). As a noble metal dopant, Pt single-atom catalysts (SACs) exhibits outstanding reactivity and selectivity, as well as excellent thermal stability and sinter resistance. The heterostructural SACs system, with an ultrahigh specific surface area (54.29 $m^2$ $g^{-1}$) and maximized catalytic active sites, exhibited outstanding sensing performance in HCHO detection (response = 33.9 at 5 ppm), high thermal stability (7.1% degradation in sensing performance after >170 h of operation at 275 °C) and good selectivity (Figure 9b,c). This enhanced performance can be mainly attributed to two major factors: the enhanced spillover effect of Pt−N/C bonding and the heterojunction formation between Pt–MCN and $SnO_2$ (Figure 9d).

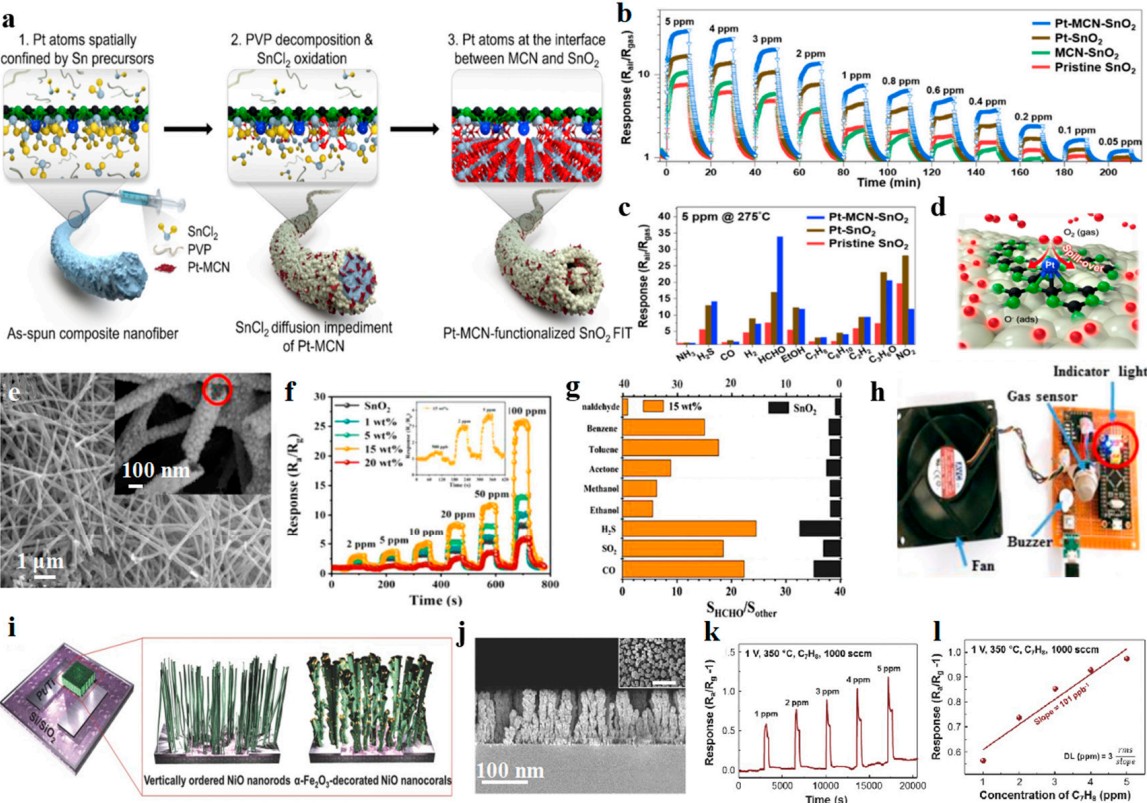

**Figure 9.** (**a**) Schematic illustration of the Pt SA delivery process and FIT structure formation; (**b**) HCHO gas sensing results of Pt–MCN–SnO$_2$ compared with other samples; (**c**) selectivity of Pt–MCN–SnO$_2$ toward different gases compared with other samples; (**d**) schematic illustration of O$_{2(gas)}$ dissociation over Pt SAs on Pt–MCN–SnO$_2$ and chemisorbed oxygen species [163]; (**e**) SEM image of 15 wt. % ZZS HNFs; (**f**) response of different ZZS HNFs-based gas sensors to different concentrations of HCHO; (**g**) the response ratios of SnO$_2$ and 15 wt. % ZZS HNFs sensors to HCHO versus other gases at 400 °C; (**h**) photographs of the lab-made detection device [96]; (**i**) schematic of vertically ordered NiO NRs on Pt-interdigitated electrodes without and with α-Fe$_2$O$_3$ decoration; (**j**) cross-sectional SEM image of the α-Fe$_2$O$_3$–NiO NRs; (**k**) responses and (**l**) calibration of responses for α-Fe$_2$O$_3$–NiO nanocorals toward 1–5 ppm Toluene [164].

Zeng et al. [165] fabricated pure In$_2$O$_3$ nanowires (PINW) and CdO–In$_2$O$_3$ composited beaded porous nanotubes (CIBPNT) by electrospinning and annealing. The CIBPNT-based sensor exhibited an excellent response (72 to 50 ppm formaldehyde) at a low operating temperature (132 °C), which is 8 times higher than PINW. The sensor based on CIBPNT showed outstanding selectivity (its responses were almost unchanged when mixed with other gases such as ethanol, methanol, acetone and toluene) and fast response/recovery time (6/12 s), and even enabled super-low-concentration (~0.1 ppm) HCHO detection. There improvements were attributed to the unique bead shape of the hollow/porous 1-D nanostructures, as well as the high catalytic activity of CdO. In addition, our group [96] also synthesized a series of MOF-derived Zn$^{2+}$-doped SnO$_2$ hollow nanofibers (ZZS HNFs) via the facile electrospinning method and annealing treatment for smart HCHO monitoring (Figure 9e). Among the as-prepared ZZS HNFs, the 15 wt. % ZZS HNFs exhibited the highest response, fastest response/recovery time (12/45 s), best selectivity and repeatability, and a detection limit as low as 500 ppb towards HCHO (Figure 9f,g). The enhancement in the sensing properties of 15 wt. % ZZS HNFs could be attributed to the high specific surface area and the increase in oxygen vacancies and chemisorbed oxygen species, benefiting from the 1-D nanostructures and Zn$^{2+}$ doping. To investigate the practical performance of the gas sensor based on 15 wt. % ZZS HNFs for monitoring HCHO, they intelligently designed a smart detection device, and realized real-time monitoring and an effective alarm process for HCHO, highlighting its great potential in smart indoor air quality monitoring (Figure 9h).

Toluene, as a volatile liquid with a special smell, is widely employed as a chemical solvent, a high-octane gasoline additive, and an important raw material for organic chemicals [166]. It is a serious hazard to the environment that can cause pollution to the air and water, due to its corrosiveness, non-degradability and bioaccumulation [167]. Bang et al. [168] synthesized Si nanowires via the metal-assisted chemical etching (MACE) method, and further deposited $TeO_2$ layers and sputtered Pt layers onto their surfaces. Under 50 ppm toluene at 200 °C, the response of the as-prepared Si NW–$TeO_2$/Pt composite (40.2) was four-fold higher than that of the Pt-lacking one. In addition, functionalization by Pt was effective in enhancing sensor response and selectivity. Pt functionalization increased the responses to $NO_2$ and toluene by 31.35% and 319.88%, respectively. The response to benzene was decreased by 9.63% after Pt functionalization. The enhanced response of Si NW–$TeO_2$/Pt was attributed to the following points: (1) the spillover effect of Pt nanoparticles; (2) the greater number of active sites generated at the interfaces of Si NWs, $TeO_2$ and Pt NPs; (3) the modulation of the potential barriers formed at the Si NWs/$TeO_2$ and $TeO_2$/Pt NPs interfaces; (4) the enhanced chemisorption and dissociation of toluene. The Si NW–$TeO_2$/Pt sensor, enabling low-temperature and low-concentration toluene detection, possesses a promising application prospect in the fields of environmental monitoring and medical diagnosis. Suh et al. [164] fabricated vertically ordered $\alpha$-$Fe_2O_3$-modified NiO NRs with a coral-like rough surface via a facile and effective glancing angle deposition (GAD) method (Figure 9i). These $\alpha$-$Fe_2O_3$-modified NiO NRs exhibited a 45.4-fold higher response to 50 ppm toluene at 350 °C compared with the bare NiO nanorods, with a good linear relationship between response and toluene concentration, demonstrating its reliability even in the monitoring of low-concentration toluene gas (1–5 ppm) (Figure 9k,l). The enhanced toluene response of $\alpha$-$Fe_2O_3$-modified NiO NRs was attributed to the unique vertically ordered nanostructure, the synergistically catalytic effects of NiO and $\alpha$-$Fe_2O_3$, and the increased potential, benefiting from the formed p–n heterojunctions, establishing its feasibility in toluene gas sensors for indoor air quality monitoring.

### 4.2. Safety Monitoring

In recent years, electric vehicle fires, gas pipeline leaks and explosions, and chemical toxic gas leaks have frequently occurred, which are almost always caused directly by, or as results of, flammable and explosive gases, such as hydrogen, CO, $H_2S$, etc. For example, lithium-ion batterers have the advantages of high voltage, light weight, high energy density, less environmental pollution, etc., and have become the first choice for electric vehicles and energy storage devices [99,169]. However, the abuse of lithium-ion batteries, such as overcharging and overdischarging, can cause irreversible oxidation–reduction reactions and release flammable gases, hence leading to safety risks [170,171]. Therefore, the real-time and accurate detection of these flammable and explosive gases based on high-performance gas sensors is considered to be an effective strategy to realize early safety warning. Resistive gas sensors based on 1-D nanomaterials, with efficient electronic transfer paths, high sensitivity, high chemical stability and especially outstanding thermal stability, have been extensively explored for monitoring these typical gases (hydrogen, CO and $H_2S$) in safety fields to prevent possible combustions and explosions [172–174].

Hydrogen is a colorless, odorless and highly flammable gas (low flammability limit of 4%) [175], which can be released under different circumstances, such as in chemical plants and from new energy batteries in thermal runaway states. Therefore, the development of rapid and highly selective sensors of hydrogen has become highly significant [176–179]. In hydrogen detection, the safety issue for the sensing materials is related to their high-performance stability in harsh environments (corrosive, high temperature, etc.), because hydrogen production and utilization always involves high temperatures. Hermawan et al. [180] prepared GaN gas sensors with a high-temperature (>400 °C) hydrogen gas-sensing property. The GaN gas sensor exhibits high responsivity (101.5 to 750 ppm hydrogen at 500 °C) and excellent maintenance properties (the fluctuation in the resistance and response values are within the error margin of 3% after 20 days at 500 °C). The XRD

pattern of GaN after sensing measurement shows no apparent change, indicating the durability and long-term thermal maintenance of the GaN sensor. By contrast, Nair et al. [172] fabricated carbon nanofibers with surface-anchored bimetallic gold–platinum nanoislands (CNFs@Au-PtNIs) via the a simple combination of electrospinning and chemical reduction methods. Benefiting from the synergistic effects of Au–Pt and CNFs, as well as its enhanced hydrogen desorption ability, hydrogen gas sensors based on CNFs@Au–PtNIs exhibited an enhanced gas-sensing performance with a response/recovery time of 6.6/18 s at room temperature. Compared with carbide-based materials, nitride-based materials had higher working temperatures and better long-term thermal maintenance, which made them suitable for hydrogen detection in harsh environments. On the other hand, carbide-based materials achieved faster responses and room-temperature operation, and can provide early warnings of hydrogen explosions in low-temperature hydrogen storage. Raza et al. [181] fabricated 1-D $SnO_2$/NiO core–shell nanowires (CSNWs) via the two-step methods of vapor–liquid–solid (VLS) deposition and atomic layer deposition (ALD) (Figure 10a). The $SnO_2$/NiO-100 (100 is the number of ALD cycles) sensor showed a high sensitivity of 114 to 500 ppm hydrogen below 500 °C, which is about 4 times higher than the pristine $SnO_2$ NWs, with an outstanding selectivity even in a complex gas environment (hydrogen, ethanol, acetone and $NH_3$) (Figure 10b,c). The enhanced hydrogen sensing performance of $SnO_2$/NiO CSNWs was attributed to its high surface to volume ratio, the p–n heterojunction that formed at the p–NiO–shell/n–$SnO_2$–core interface, and the effective optimization of the NiO shell layer.

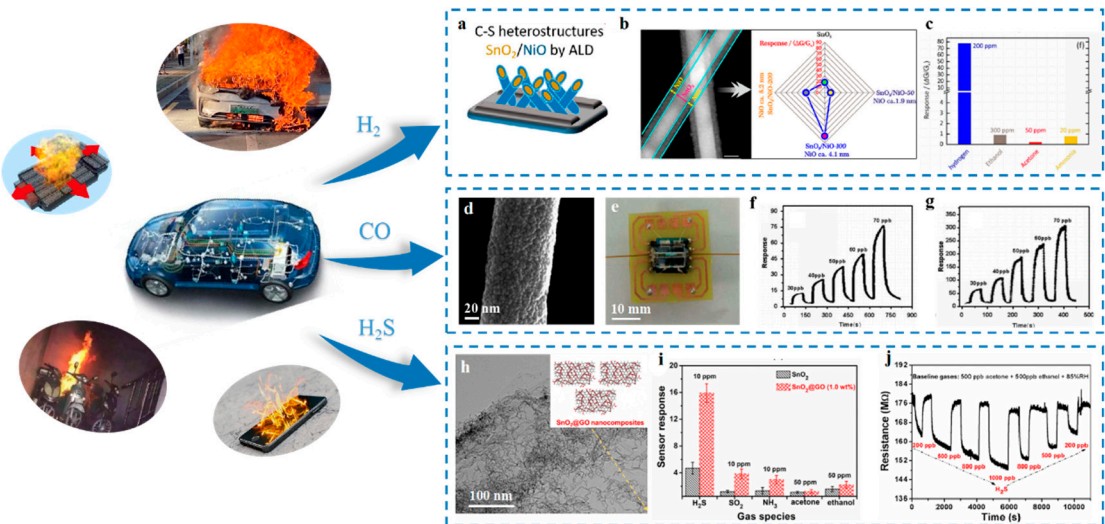

**Figure 10.** (**a**) Schematic of $SnO_2$/NiO-X CSNWs; (**b**) SEM image and sensing response of the $SnO_2$/NiO-X CSNWs heterostructures toward 200 ppm of $H_2$ at 500 °C as a function of the NiO–shell layer thickness; (**c**) response of the $SnO_2$/NiO-100 sensor toward 200 ppm $H_2$ and other interfering gases (300 ppm ethanol, 50 ppm acetone and 20 ppm $NH_3$) at 500 °C [181]; (**d**) TEM image of $TiO_2$ NFs; (**e**) photograph of the GNP–$TiO_2$ sensor; time-dependent response of (**f**) pure $TiO_2$ at 300 °C, and (**g**) GNP–$TiO_2$ at 250 °C in different CO concentrations [182]; (**h**) SEM image and schematic of $SnO_2$–GO composites; (**i**) sensing response of the $SnO_2$ gas sensor and $SnO_2$–GO gas sensor toward different gases (10 ppm $H_2S$, 10 ppm $SO_2$, 10 ppm $NH_3$, 50 ppm acetone, 50 ppm ethanol) at 70 °C; (**j**) detection of a low concentration of $H_2S$ from 200 to 1000 ppb in the mixture gases of acetone (500 ppb) and ethanol (500 ppb) based on the $SnO_2$–GO gas sensor (70 °C, RH = 85%) [183].

CO is also a highly toxic gas, and can cause fires and explosions when mixed with air under high-temperature conditions [184]. For efficient CO monitoring and early warning, Weng et al. [185] grew high-density single-crystal β-$Ga_2O_3$ nanowires on silicon (100) substrates through a vapor–liquid–solid growth method, and further compared the CO-sensing properties of pure nanowires with those of Au-modified nanowires at room temperature. The Au decorated β-$Ga_2O_3$ nanowire device achieved better results than the

pure $\beta$-$Ga_2O_3$ nanowire, with the highest sensitivity of 7.8% and a rapid response/recovery time (5.85/10.13 s) at room temperature. Kumar et al. [186] fabricated Nb-doped cryptomelane octahedral molecular sieve (Nb–OMS-2) nanofibers, and developed a reversible and selective resistive gas sensor based on nanofibers for room-temperature CO gas detection. The sensor displayed CO responses of 22% for 2 ppm and 91% for 400 ppm, respectively, with rapid response/recovery times at room temperature. It showed responses lower than 2% upon exposure to VOCs and other interfering gases (ethanol, methanol, benzene, toluene, acetone, $NH_3$, $CO_2$, etc.). The enhanced CO-sensing performance was mainly attributed to the substitution of Mn with Nb in the OMS-2 framework, the formation of the Nb–O–Mn bridge, and the reaction of CO with bridge oxygen. Wang et al. [173] developed a high-sensitivity gas sensor based on 1-D hybrid $NiO/TiO_2$ nanofibers with a modular assembly design. The ultrafine NiO nanoparticles were deposited in the surface of the $TiO_2$ nanofibers to promote gas adsorption. The experimental results showed that the unique $NiO/TiO_2$ NFs-based sensor has an excellently selective response to CO, with high sensitivity, low detection limit (1 ppm), rapid response/recovery times (<20 s) and excellent reproducibility at room temperature. The density functional theory (DFT) simulations and experimental results confirm that the selective response could be attributed to the high molecular adsorption energy of the NiO nanoparticles with 110 facets and abundant interfaces, which act synergistically to promote CO adsorption and facilitate charge transfer. Nikfarjam et al. [182] fabricated single-aligned pure $TiO_2$ NFs and gold nanoparticle (GNP)–$TiO_2$ NFs using a novel electrospinning procedure equipped with secondary electrostatic fields on very sharp triangular and rectangular electrodes for gas sensing applications (Figure 10d,e). Compared to pure $TiO_2$ NFs, GNP–$TiO_2$ NFs functioned at a lower operating temperature (250 °C), and achieved excellent sensitivity (70 to 30 ppb CO) and a low detection limit (700 ppt) (Figure 10f,g). Using GNPs as catalysts increased the sensor response by reducing the activation energy of interactions. Besides this, when GNPs came into contact with $TiO_2$ nanograins, a Schottky barrier formed between them and the electrons flowing from the $TiO_2$ nanograins to GNPs, and this further widened the depletion region in the $TiO_2$ nanograins, which is beneficial to reducing sensor conductivity.

$H_2S$ is also one of the flammable and hazardous gases that can form an explosive mixture with air and cause combustion when heated or exposed to open flames [187]. Song et al. [183] synthesized $SnO_2$ quantum wire/GO nanosheets via a simple mechanical stirring method (Figure 10h). Compared with a pure $SnO_2$ sensor, the $SnO_2$–GO sensor exhibited high sensitivity and selectivity to $H_2S$ at low temperatures (Figure 10i). More importantly, it displayed a ppb-level $H_2S$ response even in a gas mixture with $H_2S$, acetone and ethanol, with a humidity of 85% (Figure 10j). The synergetic effect between pure $SnO_2$ QWs and the GO nanosheet promoted the crucial chemical reception and transducer function, resulting in a sensitive and selective gas-sensing performance. Its room-temperature fabrication, low-temperature operation, and good compatibility with the paper substrate make this $SnO_2$–GO sensor a potential flexible gas sensor. Wang et al. [188] synthesized 1-D $ZnO/ZnSnO_3$ nanorod arrays (NRAs) with a hetero-epitaxial growth relation using ZnO nanorods as the template. The sensor based on the NRAs possessed a greatly increased quantity of active surface electrons. Hence, it exhibited a high responsivity to trace $H_2S$ gas (a detection limit of 700 ppb) at low optimum working temperatures (less than 170 °C), and a fast response/recovery time (<10 s).

### 4.3. Health Monitoring

Some gases in human breath are considered to be biomarkers of disease, so it is possible to diagnose disease by detecting these characteristic gases. For example, formaldehyde (lung cancer) [189], toluene (lung cancer) [190], $NH_3$ (hemodialysis) [191], $H_2S$ (halitosis) [192], isoprene (heart disease) [193], etc. The gas sensors employed in medical diagnosis must achieve a low power consumption, and simultaneously enable the detection of trace gases at room temperature, and even be flexible and wearable in some cases.

As a biomarker of diabetes, the acetone content in exhaled breath has been extensively studied as a non-invasive way to quantify blood glucose levels. The respiratory acetone concentration level of healthy people is generally between 0.4 and 0.9 ppm, while the concentration level of diabetic patients is >1.8 ppm. People with an acetone concentration between 0.9 and 1.8 ppm are considered at risk of developing diabetes [194,195]. Hence, it is meaningful for the development of high-sensitivity gas sensors to accurately monitor the trace-level acetone in exhaled breath. Ama et al. [196] developed a novel acetone sensor based on nanocomposites of 1-D KWO ($K_2W_7O_{22}$) nanorods/2-D $Ti_3C_2T_x$ nanosheets. This nanocomposite exhibited a high responsivity to acetone (10 times higher sensitivity comparing to a KWO-based sensor), much better humidity disturbance tolerance, and enhanced stability for months, confirming its potential use as an outstandingly sensitive and selective material for acetone detection in healthcare and the prevention of diabetes. Gong et al. [197] prepared an acetone sensor based on $\alpha$-$Fe_2O_3$/$SnO_2$ hybrid nanoarrays (HNAs) via a facile two-step chemical bath deposition method (Figure 11a–c). The $\alpha$-$Fe_2O_3$/$SnO_2$ HNAs had an enhanced sensitivity and outstanding selectivity to acetone (3.25 at 0.4 ppm), compared with those based on pure $SnO_2$ nanosheets (1.16 at 0.4 ppm) and pure $\alpha$-$Fe_2O_3$ nanorods (1.03 at 0.4 ppm) (Figure 11d,e). The enhanced performance might be attributed to the formation of a n($\alpha$-$Fe_2O_3$)-n($SnO_2$) heterostructure with 1-D/2-D hybrid architectures. The $\alpha$-$Fe_2O_3$/$SnO_2$ HNAs also possessed outstanding reproducibility, indicating their potential application in breath acetone analysis. Kim et al. [198] used a novel GO templating route combined with the electrospinning technique to synthesize highly porous Pt-functionalized $SnO_2$ flake-assembled nanofibers. This novel material exhibited high responsivity (79.4 to 1 ppm of acetone), an outstanding selectivity against eight other interfering gases (HCHO, $C_2H_5OH$, $C_2H_6S$, $H_2S$, $C_7H_8$, $NH_3$, CO, and $CH_4$), and excellent stability over 13 test cycles. This acetone sensor accurately recognized simulated diabetic breath and the breath from healthy individuals. Shingange et al. [199] synthesized 1-D porous Au-modified $LaFeO_3$ nanobelts (NBs) with a high surface area through a facile electrospinning method. The experimental results revealed that the Au/$LaFeO_3$NB-based sensor displayed an enhanced response to 125 to 40 ppm acetone and rapid response/recovery times of 26/20 s at an optimal working temperature of 100 °C, with the excellent selectivity for acetone against $NO_2$, $NH_3$, $CH_4$ and CO. Therefore, the Au/$LaFeO_3$-based sensor is regarded as a promising candidate for sensitive, ultrafast, and selective acetone detection in diabetes monitoring. Kim et al. [200] designed bimetallic nanoparticles employing bimetallic PtM (M = Pd, Rh, and Ni) nanoparticles via a protein encapsulating route supported on mesoporous $WO_3$ NFs. These structures demonstrated high sensitivity to acetone at 350 °C, and excellent selectivity toward seven interfering gases (Figure 11g,h). The sensors based on PtRh $WO_3$ NFs exhibited high sensitivity for detecting target biomarkers (even at ppb levels) in highly humid exhaled breath. Sensor arrays have been further employed to enable pattern recognition capable of discriminating between simulated biomarkers and controlled breath (Figure 11f). Wang et al. [201] fabricated poly (styrene-butadiene-styrene)/carbon nanotube (SBS/CNT) hybrid fiber using a scalable wet spinning process. The sensor based on SBS/CNT exhibited a high sensitivity (19% to 10% acetone) in a wide workable detection range, with a rapid response (<40 s) and excellent mechanical reliability. The fibers showed good deformation capabilities and a maximum deformation of 1000%. The variation in the electrical resistance of the fiber under bending is negligible. This sensor had huge potential applications in wearable and flexible electronic devices for health monitoring.

In addition to medical treatment, food safety is also a major factor related to human health. Meat, vegetables, and fruits can release characteristic gases ($NH_3$, hydrogen sulfide (discussed above) and ethanol) during the spoilage process. It is greatly necessary to detect these gases to ensure the freshness of food. Some organics will decompose to produce alcohol when food ferments. Therefore, the monitoring of alcohol has been widely applied in food safety [202]. Burris et al. [203] fabricated electrospun PANi/PEO NFs doped by different variants of graphene oxide (GO) onto interdigitated microelectrodes (IMEs) using a rotating aluminum disc collector. The sensing performance of materials can be

adjusted and enhanced by changing the reaction duration of the chemical reduced GO (crGO). At room temperature, all the three sensors exhibited a linear response and rapid response/recovery time. Sensor arrays consisting of PANi/PEO composites with thermal reduced GO (trGO), crGO-6 or crGO-24 (6 and 24 are the hours for the duration of chemical reduction) moieties successfully identified methanol, ethanol, and 1-propanol vapors using principal component analysis (PCA). These low-cost portable gas sensors can be introduced into the house or food industry to monitor the presence of VOCs. Bai et al. [204] synthesized pure and rare earth (Ce, Tm, Eu, Er, and Tb)-doped $In_2O_3$ NTs via a simple uniaxial electrospinning method. The authors found that rare earth doping can effectively improve the response and selectivity of $In_2O_3$-based gas sensors in ethanol detection. Among these as-prepared rare earth-doped $In_2O_3$ NTs, the Tb-doped $In_2O_3$ gas sensor had the highest response of 159.8 to 100 ppm ethanol at 220 °C. Its high ethanol sensitivity was mainly attributed to the following three aspects: (1) doping could greatly reduce the particle size of $In_2O_3$ NTs; (2) the distribution of three different oxygen species on the sensor surface could be controlled by Tb doping; (3) $Tb^{3+}$ cations could be oxidized to $Tb^{4+}$ cations to adsorb more oxygen ions. Compared with other $In_2O_3$-based materials with different dimensions and structures (Er/Tm/La–$In_2O_3$ 3-D ordered macroporous structures [205], Tb–$In_2O_3$ nanoparticles [206], Ce–$In_2O_3$ hierarchical flower-like microspheres [207], Dy–$In_2O_3$ nanoparticles [208] and Ce–$In_2O_3$ nanospheres [209]), these 1-D Tb–$In_2O_3$ NTs had the highest response (159.8 compared with 27.8–122) to 100 ppm ethanol and a relatively low operating temperature (220 °C compared with 175–300 °C).

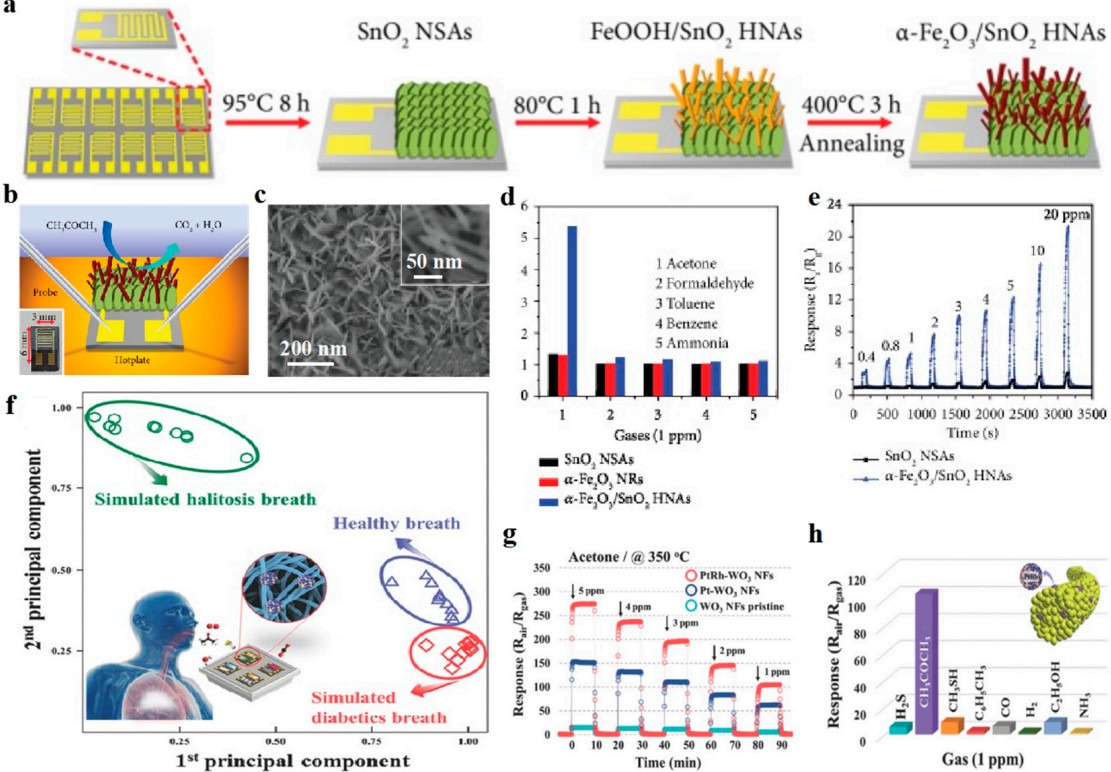

**Figure 11.** Schematic diagram of (**a**) the synthesis progress of $\alpha$-$Fe_2O_3$/$SnO_2$ HNAs and (**b**) gas sensing measurement platform; (**c**) SEM image of $\alpha$-$Fe_2O_3$/$SnO_2$ HNAs; (**d**) selectivity of the sensors to various gases at 340 °C; (**e**) the corresponding transient response curves of $SnO_2$ NSAs and $\alpha$-$Fe_2O_3$/$SnO_2$ HNAs [197]; (**f**) pattern recognition by PCA using the dataset from sensor arrays evaluating real and simulated breath. Acetone-sensing property of the PtRh-$WO_3$ NFs: (**g**) acetone (1–5 ppm)-sensing property of the PtRh–$WO_3$ NFs at 350 °C; (**h**) selective property toward 1 ppm as well as the response toward seven interfering gases [200].

The summary of resistive gas sensors based on 1-D nanomaterials toward different applications is shown in Table 1.

Table 1. Summary of resistive gas sensors based on 1-D nanomaterials toward different applications.

| Application | Target Gas | Material | Design Strategy | Performance | | | | | Ref. |
|---|---|---|---|---|---|---|---|---|---|
| | | | | Concentration | Response | $T_{response}/$ $T_{recovery}$ (s) | Temperature (°C) | Limit of Detection | |
| Environmental monitoring | NO₂ | Au–SnO₂ NFs WS₂–SiO₂ NRs ZnO NRs | Doping Composites - | 5 ppm 5 ppm 5 ppm | 180 151.2 70 | 500/223 - 16/200 | RT RT 150 | 6 ppb 13.726 ppb 1 ppm | [142] [143] [101] |
| | NH₃ | h-MoO₃ NRs | - | 5 ppm | 36 | 230/267 | 200 | - | [150] |
| | | CuPc–MOF-3 | Heterostructures | 5 ppm | 45 | - | RT | 52 ppb | [151] |
| | | AuGNR | Composites | 25 ppm | 34 | 224/178 | RT | - | [152] |
| | | Ag NC−MWCNTs | Composites | 100 ppm | 9 | 15/7 | RT | - | [144] |
| | Methanol | ZnO–NiCo₂O₄ NFs | Heterostructures | 100 ppm | 6.67 | 37/175 | 250 | - | [157] |
| | | Pd–CeO₂ NFs | Doping | 100 ppm | 6.95 | - | 200 | 402 ppb | [158] |
| | | CNT–ZnO | Composites | 25 ppm | 72.6 | 3.15/3.45 | 250 | - | [159] |
| | HCHO | Pt–MCN–SnO₂ | Doping/Heterostructures | 5 ppm | 33.9 | - | 275 | 50 ppb | [163] |
| | | ZZS HNFs | Doping | 100 ppm | 25.7 | 12/45 | 400 | 500 ppb | [96] |
| | | CdO–In₂O₃ NTs | Heterostructures | 50 ppm | 72 | 6/12 | 132 | 100 ppb | [165] |
| | Toluene | Pt–TeO₂–Si NW | Doping/Composites | 50 ppm | 45 | 20/500 | 200 | - | [168] |
| | | α-Fe₂O₃–NiO nanocorals | Heterostructures | 50 ppm | 45.4 | - | 350 | 22 ppb | [164] |
| Safety monitoring | H₂ | Au-Pt-CNFs | Doping/Composites | 500 ppm | 33 | 6.6/18 | RT | - | [172] |
| | | SnO₂/NiO CSNWs | Heterostructures | 500 ppm | 114 | 120/660 | 500 | 0.9 ppm | [181] |
| | CO | Au–β–Ga₂O₃ NWs | Doping | 100 ppm | 4.8 | 21.14/21.34 | RT | - | [185] |
| | | Nb–OMS–2 NFs | Doping | 2 ppm | 22 | 25/40 | RT | - | [186] |
| | | NiO–TiO₂ HNFs | Heterostructures | 50 ppm | 2.07 | 10/20 | RT | 1 ppm | [173] |
| | | GNP–TiO₂ NFs | Doping | 30 ppb | 75 | 3/4 | 250 | 700 ppt | [182] |
| | H₂S | SnO₂–GO | Composites | 10 ppm | 17.9 | 12/137 | 70 | 61 ppb | [177] |
| | | ZnO–ZnSnO₃ NRs | Heterostructures | 30 ppm | 137.9 | 14/26 | 165 | 700 ppb | [188] |

**Table 1.** *Cont.*

| Application | Target Gas | Material | Design Strategy | Performance | | | | | Ref. |
|---|---|---|---|---|---|---|---|---|---|
| | | | | Concentration | Response | $T_{response}/$ $T_{recovery}$ (s) | Temperature (°C) | Limit of Detection | |
| Health monitoring | Acetone | KWO–Ti$_3$C$_2$T$_X$ | Composites | 2.86 ppm | 2.5 | - | RT | - | [196] |
| | | α-Fe$_2$O$_3$–SnO$_2$ | Heterostructures | 1 ppm | 5.37 | 14/70 | 340 | 0.4 ppm | [197] |
| | | Pt–SnO$_2$ NFs | Doping | 5 ppm | 245.2 | 12.7/- | 350 | 100 ppb | [198] |
| | | Au–LaFeO$_3$ NBs | Doping | 40 ppm | 125 | 26/20 | 100 | 267 ppb | [199] |
| | | PtPd–WO$_3$ NFs | Doping | 1 ppm | 97.5 | 4.2/204 | 300 | 1.07 ppb | [200] |
| | Ethanol | GO–PANi–PEO | Composites | 200 ppm | 2.1 | 30/252 | RT | 15 ppm | [203] |
| | | Tb–In$_2$O$_3$ NTs | Doping | 100 ppm | 159.8 | 1/60 | 220 | - | [204] |

## 5. Summary and Perspective

In this review, the design and optimization strategies for 1-D nanomaterials and their potential use in resistive gas sensors were discussed in detail. Resistive gas sensors, as the dominant sensors in gas detection, possess the advantages of rapid responsivity, outstanding sensitivity, excellent repeatability and low costs. However, they also suffer from poor selectivity and high operating temperatures. At present, to achieve the best gas sensing performance, there are three main ways to design and optimize gas-sensitive materials: doping, heterostructures and composites. These methods can change the grain size, porosity and specific surface area of the material, improve the electron transport characteristics, and increase the surface adsorption active sites, thereby improving the sensitivity and selectivity of gas sensors.

In addition, the application of resistive gas sensors based on 1-D nanomaterials can be mainly summarized into three categories: environmental monitoring, safety monitoring, and health monitoring. Although the applicability of the sensor in these fields has made some progress, the following problems still need to be solved: (1) For novel composite materials prepared from more than two types of materials, it is necessary to strengthen the research on their microstructure and synergistic effects. (2) Resistive gas-sensing materials have high impedance at room temperature, and the change in conductivity caused by gas adsorption is extremely insignificant. As a result, the energy consumption of the resistive gas sensor is relatively high. In order to reduce the working temperature of the resistive gas sensor, it is necessary to innovatively develop high-performance low-temperature gas sensing materials and further clarify their working mechanism. (3) Selectivity is a critical parameter of gas sensors. Unfortunately, the causes of selective behavior are not completely understood to date. As such, it is essential to investigate the surface reaction processes by measuring the reaction products emanating from the sensor surface using various techniques such as gas chromatography and DRIFT. (4) Gas monitoring is generally employed in complex gas environments, and multiple gases need to be detected at the same time. Therefore, the development of the electronic nose (e-nose), a gas recognition system based on integrated gas sensor arrays and multi-sensor information fusion technology, is of great significance. The e-nose has the characteristics of portability; real-time, online, in-situ analysis; etc., enabling odor identification, gas concentration identification in complex environments, and the identification of combustible gases, organic volatiles or toxic gases. (5) With the development of the information age, flexible, wearable and intelligent sensors will become the inevitable trend, with both opportunities and challenges.

**Author Contributions:** Conceptualization, J.W. and L.Z.; writing—original draft preparation, Z.W.; writing—review and editing, J.W., L.Z. and S.S.; supervision, W.Y. and J.W.; funding acquisition, J.W. All authors have read and agreed to the published version of the manuscript.

**Funding:** This research was funded by the National Natural Science Foundation of China, grant number 51803164; Basic and Public Projects of Zhejiang Province, grant number LGF21E020001; China Postdoctoral Science Foundation, grant number 2020M683467; Fundamental Research Foundation for the Central Universities of China, grant number xjh012020031.

**Institutional Review Board Statement:** Not applicable.

**Informed Consent Statement:** Not applicable.

**Data Availability Statement:** Not applicable.

**Acknowledgments:** We thank the financial support from the National Natural Science Foundation of China (51803164), Basic and Public Projects of Zhejiang Province (LGF21E020001), China Postdoctoral Science Foundation (2020M683467), and the Fundamental Research Foundation for the Central Universities of China (xjh012020031).

**Conflicts of Interest:** The authors declare no conflict of interest.

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
