# Peer review of "One-Dimensional Nanomaterials in Resistive Gas Sensor: From Material Design to Application"

_chemosensors, doi:10.3390/chemosensors9080198_

Round 1

Reviewer 1 Report

Comments to the author:

In this manuscript, the authors presented the state of the art in the design and optimization strategies of 1-D nanomaterials and its potential as resistive gas sensors. The authors prepared a very didactic manuscript starting by the description of the key performance indicators of the gas sensors, the different configurations, sensing mechanisms and then giving examples the most outperforming works found in literature divided by application. Overall, I would recommend the publication of this contribution.

The following are some questions and suggestions for improving their work:

Major issues:

  • Figure 1 is not clear very clear; the authors are comparing techniques but for which kind of materials? The authors claimed that the main disadvantage of resistive sensors is their poor selectivity and that could be solved with 1D materials, but which kind of materials were used for this plot? Also, the reason that 1D materials will provide selectivity is not clear.
  • The section ‘2.3 Sensing mechanism of resistive gas sensor’ should be improved. The authors just described the case of metal oxide semiconductors, what about the other kind of 1D materials? Are the mechanisms the same? Also, with this mechanisms selectivity is not understood. If the mechanism is related to the amount of electron acceptors and donors then I do not understand why this will drive any selectivity to a certain single gas.
  • The section ‘2.4 1-D nanomaterials’ should be improved. The different kind of 1D materials should be properly described as the different families. Also figure 6 is a very specific case of SiO2 nanowires.
  • Regarding the section ‘3. Materials design’, the use od ad hoc (supra)molecular receptors of the analyte of interest has not been considered, only pristine materials. Could the authors comment on that?
  • In all the examples cited in the application section a more detailed discussion is expected. The authors should comment of the selectivity of the materials they have chosen, if they refer to a single or a mixture of gases and explain if the authors have tested their sensors in the presence of interfering gases. Besides, if different materials have been used for the detection of a specific gas, a comparison of their performance is expected. If the main advantage of using 1D materials is the selectivity, this parameter should be discussed in detail in every case.

Minor issues:

  • When 0D materials are deposited on a platform, the aggregation is no longer a problem. The authors should mention that.
  • In the section ‘2.1 Performance of resistive gas sensor’ could the authors explain better which is the difference between stability and maintenance?
  • Figure 1 quality should be improved. Many of the titles can be barely seen. Also it is difficult to understand where the sensing materials are placed.

Reviewer 2 Report

The authors have presented a very elaborated literature survey in their review titled "One-Dimensional Nanomaterials in Resistive Gas Sensor: Material Design to Application". The review presents a comprehensive discussion of application of one-dimensional nanomaterials for gas sensor application. The introduction is also nicely written. Moreover, resistive gas sensing mechanism and configuration are addressed, and the materials designs and strategies to improve the gas sensing mechanism are highlighted.  It will be a very good guideline for researchers in this area. Recommend accepting for publication in chemosensors with major revision following the below concerns:

  1. According to the author’s claim, the nanomaterials can be divided based on their dimensionality and 1D material has a high length-to-diameter ratio, more exposed active sites, and large surface area compared to other nano dimensional materials. For the first reason, I do well recognize, however, I don’t agree with the latter claims, as some of the 3D materials with hierarchical nanostructures such as SnO2 microsphere exhibited large porosity and high specific surface area more than 140 m2/g (doi.org/10.1016/j.ceramint.2019.05.043), or some MOFs (doi.org/10.1039/C5CS00040H) and 3D graphene frameworks (10.3390/s21103386) have also similar properties exceeding some of 1D. Therefore, I suggest the author write a more comprehensive discussion on 3D nano/microstructures and include those references.

  2. The construction of heterostructures is indeed one of the best approaches in effectively enhancing the MOSs nanostructures and the reasons have been well-written by the authors. However, we should note that heterostructures design in some cases led to the increase of operating temperature due to higher activation barrier at the interface (10.3390/nano11041026 , doi.org/10.1016/j.jmst.2020.02.041, etc). Writing the disadvantages of doping, heterostructures and composites are highly recommended, given the evidence of the suggested references.

  1. 1D materials have been applied for safety monitoring, especially for flammable and explosive gas detection. In particular to hydrogen detection, the safety issue for the sensing materials is that their high-performance stability in the harsh environment (corrosive, high temperature, etc) because hydrogen production and utilization always involve high temperature. This is in line with author's scenario in which hydrogen is used in chemical plants and the new-energy batteries under a thermal runaway state. The example of materials are somewhat not appropriate, e.g. mentioning carbon nanofibers (CNFs) which cant withstand at higher temperature. Moreover, metal nitride in a recent case has shown high stability for hydrogen detection at 400oC ( doi.org/10.1016/j.snb.2018.08.021 )which in some extent better than any 1D carbon-based material. The author is suggested to revise the example of CNFs and includes the comparison with nitride or carbide-based material.

  1. 1D material is beneficial for health monitoring as it can easily be embedded in flexible electronics and sensor devices. The author is recommended to write the mechanical property of 1D materials in relation to sensors for health monitoring.

    5. Authors should also provide comparative details of different nano dimensions used for the same applications that may be superior or inferior to 1D materials in their performance.

Reviewer 3 Report

one-dimensional nanomaterials hold great promise for electronic devices. The authors presented a thorough overview of the recent advances in utilization of 1D nanomaterials for resistive gas sensor. Specifically authors summarized the three strategies including metal doping, fabrication of heterostructure and synthesis of nanocomposites to design efficient materials for multiple applications. The topic of this review is quite interesting and fall in the scope of the journal Chemosensors. In the reviewer’s opinion, this review is publishable after addressing the following points.

  1. In the opening sentence of abstract, serious or series? Please check.
  2. Figure 1 is interesting to see the advantages/disadvantages of various sensors. Please provide references to the panel figures.
  3. The reviewer agrees that sensitivity, selectivity, stability, speed (response-recovery), i.e. “4S” is very important parameters of chemical gas sensors (Adv. Mater. 2016, 28, 795–831). The discussion of the materials design strategies can be more insightful to show how to design specific 1D materials to optimize these parameters.
  4. For gas sensors based on metal oxides, the selectivity and operating temperature have been of great concern. Especially, room or low temperature sensing has attracted enormous interest to realize a low power consumption. See some recent reports: Chemosensors 2020, 8, 72; Sensors Actuators: B. Chemical 2021, 341, 129919; ACS Appl. Mater. Interfaces 2020, 12, 20704; Microchemical Journal 2021, 165, 106111; Journal of Hazardous Materials 2021, 416, 125830; Journal of Hazardous Materials 411 (2021) 125120. Authors are better to discuss how to lower the operating temperatures through engineering 1D materials.
  5. For metal doping, Ni doesn’t belong to the noble metals. Please correct. Noble functionalization on metal oxides can also lower the operating temperature of gas sensors, see some recent reports Mater. Horiz., 2020, 7, 1519; ACS Applied Nano Materials ‏ 2021, 4, 7-12; Sensors Actuators: B. Chemical 2021,‏ 333, 129545; Sensors Actuators: B. Chemical 307 (2020) 127616; ACS Applied Nano Materials 2021, 4, 2752; ‏ Sensors Actuators: B. Chemical 2021, 331, 129425; ACS Appl. Mater. Interfaces 2020, 12, 46267. In addition, discussion on the sensing mechanism of noble metals can be more informative based on the single atom Pt catalyst.
  6. For heterojunctions, the n-n junctions might result in increase of the materials resistance as recently reported in Sensors Actuators: B. Chemical 310 (2020) 127846.
  7. In page 10, the formation of heterojunctions can not only effectively accelerate the electron transport, but also enhance the oxygen adsorption and form abundant oxygen vacancies. Please give some references to these claims.
  8. Quality of some figures should be improved, for example, Fig 3a, 3e and 3f…
  9. Finally what is the weakness of 1D materials? For example, the integration of 1D materials into a sensor device is not discussed. For sensor micro-fabrication, the in-situ deposited 2D MOS thin films using techniques like chemical deposition, evaporation or sputtering are more advantageous, eg. Journal of Hazardous Materials 411 (2021) 125120; Microchemical Journal 2021, 165, 106111; Sensors Actuators B: Chemical 2021, 329, 129169; CrystEngComm, 2021, 23, 3654 – 3663; Sensors Actuators B: Chemical 2021, 317, 128217; Sensors Actuators B: Chemical 2021, 341, 129996; Sensors 2018, 18, 4177; Sensors 2018, 3, 735.

Round 2

Reviewer 1 Report

The authors have answered properly all the questions I have raised. The paper is now ready for publication.

Reviewer 2 Report

The authors have addressed all concerns raised by the review, which significantly improved the manuscript quality. Thus, my recommendation is to publish in the present from.